



# The Impact of Volcanic Eruptions of Different Magnitude on Stratospheric Water Vapour in the Tropics

Clarissa Alicia Kroll[1], Sally Dacie[1], Alon Azoulay[1,2], Hauke Schmidt[1], and Claudia Timmreck[1]

[1]Max Planck Institute for Meteorology, Hamburg, Germany
[2]now at: Remote Sensing Technology Institute (IMF), German Aerospace Center (DLR), Oberpfaffenhofen, Germany

**Correspondence:** Clarissa Kroll (clarissa.kroll@mpimet.mpg.de)

**Abstract.** Volcanic eruptions increase the stratospheric water vapour (SWV) entry via long wave heating through the aerosol layer in the cold point region, and this additional SWV alters the atmospheric energy budget. We analyze tropical volcanic eruptions of different eruption strengths with sulfur (S) injections ranging from 2.5 Tg S up to 40 Tg S using EVAens, the 100-member ensemble of the Max Planck Institute - Earth System Model in its low resolution configuration (MPI-ESM-LR) with artificial volcanic forcing generated by the Easy Volcanic Aerosol (EVA) tool. Significant increases in SWV are found for the mean over all ensemble members from 2.5 Tg S onward ranging between [5,160] %, while for single ensemble members the standard deviation between the control run members (0 Tg S) is larger than SWV increase of single ensemble members for the eruption strengths up to 20 Tg S. A historical simulation using observation based forcing files of the Mt. Pinatubo eruption, which was estimated to have emitted $(7.5 \pm 2.5)$ Tg S, returns SWV increases slightly higher than the 10 Tg S EVAens simulations due to differences in the aerosol profile shape. An additional amplification of the tape recorder signal is also apparent, which is not present in the 10 Tg S run. These differences underline that it is not only the eruption volume, but also the aerosol layer shape and location with respect to the cold point that have to be considered for post-eruption SWV increases. The additional tropical clear sky SWV forcing for the different eruption strengths amounts to [0.02, 0.65] W/m$^2$, ranging between [2.5, 4] percent of the aerosol radiative forcing in the 10 Tg S scenario. The monthly cold point temperature increases leading to the SWV increase are not linear with respect to AOD nor is the corresponding SWV forcing, among others, due to hysteresis effects, seasonal dependencies, aerosol profile heights, and feedbacks. However, knowledge of the cold point temperature increase allows for an estimation of SWV increases with a 12 % increase per Kelvin increase in mean cold point temperature, and yearly averages show an approximately linear behaviour in the cold point warming and SWV forcing with respect to the AOD.

## 1 Introduction

It has been established that the entry of water vapour into the stratosphere is largely controlled by the temperature of the tropical tropopause (e.g, Brewer, 1949; Mote et al., 1996; Fueglistaler et al., 2009; Dessler et al., 2014). Following up on the discussion of the long term increasing trend in stratospheric water vapor (SWV) observed during the 1980s and 1990s it was proposed that volcanic eruptions could be influencing the SWV budget (e.g, Rosenlof et al., 2001; Joshi and Shine, 2003). Mainly two





processes are considered: the direct injection from the volcanic plume, and the indirect mechanism due to an increase of the tropopause temperature. The increased SWV levels may remain in the stratosphere for more than 5 years (Hall and Waugh, 1997), even though the volcanic aerosols are sedimenting out of the stratosphere within 1 - 3 years (Robock, 2000). However, the magnitude of the SWV increase and the contribution from the different entry mechanisms are still unclear.

80 % of the eruption material is water vapour (Coffey, 1996), which could be directly injected into the stratosphere during an eruption event. But although the SWV originating form the direct injection can be detected shortly after the eruption event, it is a singular event and the corresponding elevated SWV levels are spread in the stratosphere and are not distinguishable from the background SWV anymore. Satellite evidence for the direct injection events exist and is discussed briefly by Schwartz et al. (Schwartz et al. (2013)) and in depth by Sioris et al. (Sioris et al. (2016a), Sioris et al. (2016b)). Based on a model study

Joshi and Jones (2009), hypothesized that the environment surrounding the plume can also have a significant impact on the amount of SWV injected directly.

This study will focus on the indirect entry mechanism. In contrast to the direct entry, it can act for months or even years after volcanic eruptions since it depends on the aerosol layer in the stratosphere, and not on the eruption event itself. It is caused by

the long wave (LW) heating by the aerosol layer, which leads to increased cold point temperatures. Consequently the saturation water vapour pressure at the cold point is increased, reducing the "freeze trap" effect originating from the increasingly low temperatures and consequent loss of WV due to ice formation and fallout. The reduced freezing trap character enhances the entry of water vapour into the stratosphere.

In an early, idealized study Joshi and Shine (2003) already underlined the importance of the aerosol profile and corresponding

LW-heating in the tropopause region. Despite the mechanisms being known, the analysis of the indirect entry mechanism is still complicated by scarce observational data, since the aerosols have to be in the stratosphere to open the indirect pathway and even if the plume reaches the stratosphere, the amount of sulfur for aerosol formation may be too low, leading to a signal obscured by internal variability. If a signal can be observed however, it is only one individual event occurring on the background of natural variability. An assessment of how typical the respective event is, makes a larger amount of data for similar events

or ensemble simulations necessary. Additionally, volcanic eruptions fulfilling the criteria needed to open the indirect pathway may lead to retrieval problems or outages of observational instruments as was the case for the most important eruption of the last century, Mt. Pinatubo. Even if SWV increases were recorded, as was the case for Mt. Pinatubo by SAGE II, the data usage is discouraged as discrepancies between different satellites can not be satisfactory explained (Fueglistaler et al., 2013).

The scarcity of observational data is also reflected in the quality of the available reanalysis products for SWV, whose usage in

general is discouraged in some papers (e.g, Davis et al., 2017) and which sometimes do not implicitly account for the volcanic forcing at all (Diallo et al. (2017), Tao et al. (2019)). The latter problem was also found by Löffler et al. (2016) when comparing their simulated SWV increases after the eruption of Mt. Pinatubo with ERAinterim reanalysis. Nevertheless by performing a regression analysis of water vapour entering the stratosphere as simulated by a trajectory model fed by reanalysis input, Dessler et al. (2014) found a SWV peak partially overlapping with the aerosol optical depth (AOD) signal of Mt. Pinatubo.



As the SWV increase occurred before the eruption and AOD increase, the question remained if the peak in the residual might instead be caused by another source of variability. Another possible issue in the analysis was that some of the effects modeled by the regressors are themselves influenced by volcanic eruptions, which may lead to the volcanic signal being attributed to a different mechanism leading to the SWV increase like increases of the Brewer Dobson circulation. In the case of a volcanic eruption these increases in the Brewer Dobson circulation are caused by the LW heating due to the aerosol layer and should be

attributed to the volcanic signal consequently. Tao et al. (2019) also undertook an indirect quantification of the SWV increase after volcanic eruption via another regression analysis. While using a Lagrangian model fed with different reanalysis sources, they explicitly accounted for volcanic source terms. They found a clear volcanic signal in the expected time frame, but the magnitude of the SWV increase was highly variable between the different reanalysis data sources.

After entering the stratosphere, the additional SWV affects both stratospheric chemistry with respect to ozone loss (Robrecht et al. (2019), Rosenlof (2018), Tian et al. (2009)) and $SO_2$ oxidation (Bekki (1995)) as well as the radiative budget of the entire atmosphere (Solomon et al., 2010). Despite the forcing originating from the additional SWV often being mentioned as a motivation for studies, few studies exist on the forcing effect of the SWV increase after volcanic eruptions. Independent of volcanic eruptions, Forster and Shine (2002) analysed the forcing impact of SWV changes in an artificial SWV profile. In a

study focusing on the Mt. Pinatubo eruption, Joshi and Shine (2003) calculated the additional global forcing originating from the post eruption SWV increase. As this was a side study of their paper, they did not investigate the temporal evolution nor the impact of different eruption strengths. In a study on the direct injection, Joshi and Jones (2009) quantified the LW-component of the SWV forcing indirectly, but their setup did not allow for a quantification of the additional contribution of the indirect pathway. Most recently Krishnamohan et al. (2019) attributed changes in their TOA imbalances for different geoengineering

scenarios to a large influence of SWV. However, they did not separate the contributions of aerosol forcing and SWV.

So far, the question remains open what the critical magnitude is for an eruption to have a significant impact on SWV content, what the radiative consequences of the SWV increase are and if these effects can be predicted based on information of the eruption magnitude or AOD. In this study we therefore investigate the changes in stratospheric water vapour originating from

the indirect pathway using a large ensemble of coupled climate model simulations with 100 ensemble members each for five eruption strengths described by changing amount of stratospheric sulfur and a control run, called the EVAens (Azoulay et al. *in preparation*, 2020). The idealized setup using forcing files generated with the Easy Volcanic Aerosol tool (EVA) offers the unique opportunity of a direct comparison between the different eruption strengths since the date and location of the eruptions are identical and all ensembles have the same set of starting conditions. By comparing with the control run a direct

quantification of the SWV increase is possible. The large ensemble size allows us to perform an analysis of the sensitivity of the increase in stratospheric water vapour to the eruption strength along with its statistical significance. The critical eruption strengths that cause stratospheric water vapour perturbations beyond the internal variability of the model are identified when analyzing the individual ensemble members.





In the following Sect. 2 the model setup of our study and the forcing calculations are described. In Sect. 3 we present results

starting with the top of the atmosphere imbalances and the changes in atmospheric temperature profiles. We put a particular emphasis on the changes in the annual cycle of vertically propagating water vapor signals and the intra-ensemble variability. With a comparison to a historical simulation of Mt. Pinatubo we highlight the importance of the shape and position of the aerosol layer for SWV entry. Finally we discuss the determination of the stratospherically adjusted forcing caused by the additional water vapour. Sect. 4 is used for discussion of our results in context of earlier studies on stratospheric water vapour

changes. Our main findings are summarized in Sect. 5 which also gives an outlook to possible further studies.

## 2  Methods

### 2.1  The EVAens ensemble and the GE historical simulations

This study is based on two sets of large ensemble simulations both covering the time frame of January 1991 to December 1993

but using different volcanic forcing data sets: the EVAens (Azoulay et al. *in preparation*, 2020) and a subset of the Max Planck Institute Grand Ensemble (MPI-GE) historical simulations (Maher et al., 2019).

Both ensemble simulations are performed with the Max Planck Institute - Earth System Model in its low resolution configuration (MPI-ESM-LR) (version, MPI-ESM 1.1.00p2). We apply an intermediate model version between the CMIP5 version (Giorgetta et al., 2013) and the CMIP6 version (Mauritsen et al. (2019)) of the MPI-ESM, with a behaviour more similar to

the CMIP6 version. The MPI-ESM itself is a coupled model including the atmosphere component ECHAM (version echam-6.3.01p3, Stevens and Bony (2013)), the land component JSBACH (version jsbach-3.00, Reick et al. (2013), Schneck et al. (2013)), the ocean component MPIOM (version mpiom-1.6.1p1, Marsland et al. (2003), Jungclaus et al. (2013)), and the biogeochemistry component HAMOCC (HAMOCC5.2, Ilyina et al. (2013)).

In the model setup the atmosphere is run in a T63L47 configuration corresponding to a horizontal resolution of about 1.9 ° with

47 pressure levels up to 0.01 hPa. The influence of the sponge layer in the uppermost model layer reaches down to a height of 65 km with continuously decreasing impact. As the MPI-ESM-LR does not include interactive atmospheric chemistry or aerosols, the volcanic aerosols are prescribed by monthly and zonal mean values of their optical properties - the extinction, single scattering albedo, and asymmetry factor - in 16 long wave and 14 short wave bands. During the simulation these monthly aerosol properties are interpolated linearly in time.

In the MPI-GE historical simulations the optical properties of the volcanic aerosols are prescribed with an updated version of the PADS data set (Stenchikov et al. (1998), Driscoll et al. (2012), Schmidt et al. (2013)). The Mt. Pinatubo eruption in June 1991 lies in the investigated time frame.

In the EVAens the setup of the MPI-GE historical simulations was kept, changing only the representation of stratospheric aerosols. The respective forcing files were generated using the Easy Volcanic Aerosol (EVA) forcing generator (Toohey et al.

(2016)). Six 100 member ensembles are created: A control run with zero sulfur emission only considering the EVA background aerosol and five volcanic eruption runs (2.5 Tg S, 5 Tg S, 10 Tg S, 20 Tg S, 40 Tg S) with EVA background aerosol and with





the volcanic eruptions occurring in June 1991 at the equator are considered, each with 100 ensemble members. Each of the these 100 ensemble members was started from one of the different and independent runs of the MPI-GE historical simulations in 1991 (Maher et al. (2019)). Beside the volcanic aerosols, the forcing files include an aerosol background for the industrial-
ized period supplied by EVA.

The aerosol optical depths (AOD) for the five different eruption strengths of the EVA forcing along with the PADS Mt. Pinatubo forcing are shown in Fig. 1 for the 550 nm waveband. All EVA aerosol distributions have very similar patterns, but differ in magnitude and the duration of elevated AOD levels. Whereas the 2.5 Tg S run returns close to background conditions
within 3.5 years after the eruption, the 40 Tg S run only declines to the peak values of the 2.5 Tg S run within this time. The PADS data set has a higher background AOD level than the EVA data sets in the months before the eruption. With a sulfur amount of $(7.5 \pm 2.5)$ Tg (Timmreck et al., 2018) the Mt. Pinatubo AOD should be comparable to the 5 Tg and 10 Tg EVA data set. Generally, the AOD in the PADS data set does not spread as fast to higher latitudes after the eruption. Additionally, the AOD values tend to be slightly higher than the values for the 10 Tg EVA data set and persist for a longer time at elevated
levels.

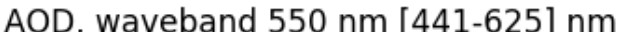

**Figure 1.** Aerosol optical depth (AOD) for the five volcanically perturbed EVAens runs (2.5 Tg S, 5 Tg S, 10 Tg S, 20 Tg S and 40 Tg S) and the PADS Mt. Pinatubo compilation. The time evolution of the zonal average AOD is shown for all latitudes considering the 550 nm waveband (441 nm - 625 nm).)



As we merely prescribe the aerosols only the indirect pathway and not the direct injection is simulated in the EVAens. In the following work all anomalies will be defined as the value difference between the volcanically perturbed (P) and unperturbed 0 Tg S (U) ensemble means (P-U).

### 2.2 Stratospherically adjusted clear sky forcing calculations

The stratospherically adjusted clear sky radiative forcing originating from the increase of stratospheric water vapour due to the indirect pathway (i.e. via tropopause warming by the aerosol layer) is calculated using the 1D radiative convective equilibirum (RCE) model konrad (Kluft et al. (2019), Dacie et al. (2019)). Konrad is designed to represent the tropical atmosphere. It uses the Rapid Radiative Transfer Model for GCMs (RRTMG) and a simple convective adjustment that fixes tropospheric temperatures according to a moist adiabat.

For each eruption strength, including the 0 Tg S eruption, the ensemble mean of the clear sky humidity profile in the tropical region [-5,5]° latitude is determined from the EVAens output. In order to compute the adjusted radiative forcing due to the increased SWV, the difference between the fluxes in an equilibrated atmosphere with and without the additional SWV must be calculated. To determine the equilibrated reference without additional SWV konrad is run to equilibrium with the humidity profile and chemical composition of the atmosphere fixed to the values from the 0 Tg S runs. The final surface temperature

lies between [298-301] K. Starting from this equilibrium state, only the SWV profile is replaced by the volcanically perturbed EVAens humidity profile and a new equilibrium is calculated while keeping the surface temperature fixed, but allowing for the temperature above the convective top to adjust. Using both equilibrium states the adjusted SWV forcing is determined. The corresponding instantaneous forcing can be calculated by calculating the flux changes without running the perturbed atmosphere into equilibrium.

For the all sky case the contribution of clouds is investigated by additionally taking a 20 % fraction of high level clouds between 200 hPa and 300 hPa into consideration, while low levels clouds are considered in the albedo settings.

In order to relate the SWV forcing to the aerosol forcing, the instantaneous aerosol forcing is calculated using a double radiation call in the MPI-ESM. For the double radiation call fluxes are calculated for each time step once using the atmospheric

conditions with and without aerosol. The stratospheric background aerosol is corrected for by additionally calculating the forcing for the corresponding 0 Tg S run and subtracting it.





## 3 Results

### 3.1 Effects on the time evolution of TOA radiative imbalance and surface temperature

In order to connect the amount of emitted sulfur to its impact on the energy budget of the system, we analyze the top of the atmosphere (TOA) radiative imbalance after the volcanic eruptions as well as changes in surface temperature. The era of negative global radiative TOA imbalance after the volcanic eruptions during which more energy leaves the earth-atmosphere system than is taken up lasts between 17 and 28 months (Fig. 2). For the lower emissions (2.5 Tg S - 5 Tg S) the standard error of the negative TOA imbalance permanently overlaps with zero imbalance. Consequently single ensemble members of the 0 Tg S can produce similar signals as these lower emission run due to internal variability. Overall the TOA imbalance exhibits a roughly linear relationship with respect to the emitted sulfur mass (see Fig. A).

As a consequence of the negative TOA imbalance the surface temperatures of the volcanically perturbed runs decrease. The range of ensemble mean global temperature decrease for the EVAens is -[0.09,1.30] K, when it is at its maximum (Fig. 2). As the surface temperature follows the TOA imbalance, the change in surface temperature is also roughly linear with respect to the emitted S mass (Fig. A).

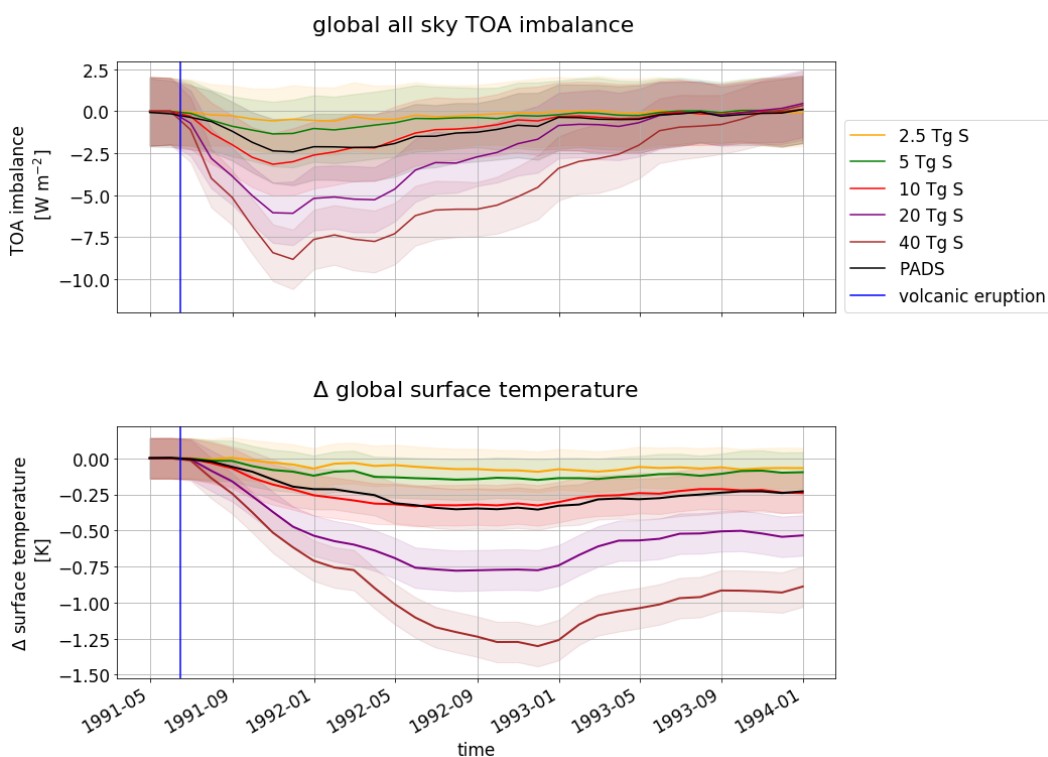

**Figure 2.** Global top of the atmosphere (TOA) radiative imbalance and anomaly of surface temperature in the five volcanically perturbed EVAens runs (2.5 Tg S, 5 Tg S, 10 Tg S, 20 Tg S and 40 Tg S). For each run the standard errors of the mean are shown. The vertical blue line marks the eruption time. The plots also show the values for the MPI-GE historical simulations (PADS).

## 3.2 Effects on the cold point temperature

The effect of the volcanic forcing on the inner tropical temperature profile is visualized in Fig. 3 showing the EVAens ensemble mean of the temperature profiles three months after the eruption. The month of September was chosen as an example since it lies in the time frame within which the annual cycle of water vapour entry into the stratosphere is enhanced. Due to the location

of the aerosol peak above the cold point[1] the largest temperature changes occur in the lower stratosphere with increases up to 24 K in the 40 Tg S ensemble. The cold point warming reaches maximum values of 8 K in the case of the 40 Tg S run. Also visible in the figure is the downwards shift of the cold point with increasing sulfur burden caused by the stratospheric warming leading to a downwards shift in the cold point levels. This effect is amplified by tropospheric cooling (compare surface cooling in Fig. 2) due to the back scattering of solar radiation through the volcanic aerosols.


---

[1]Point of lowest temperature between troposphere and stratosphere.



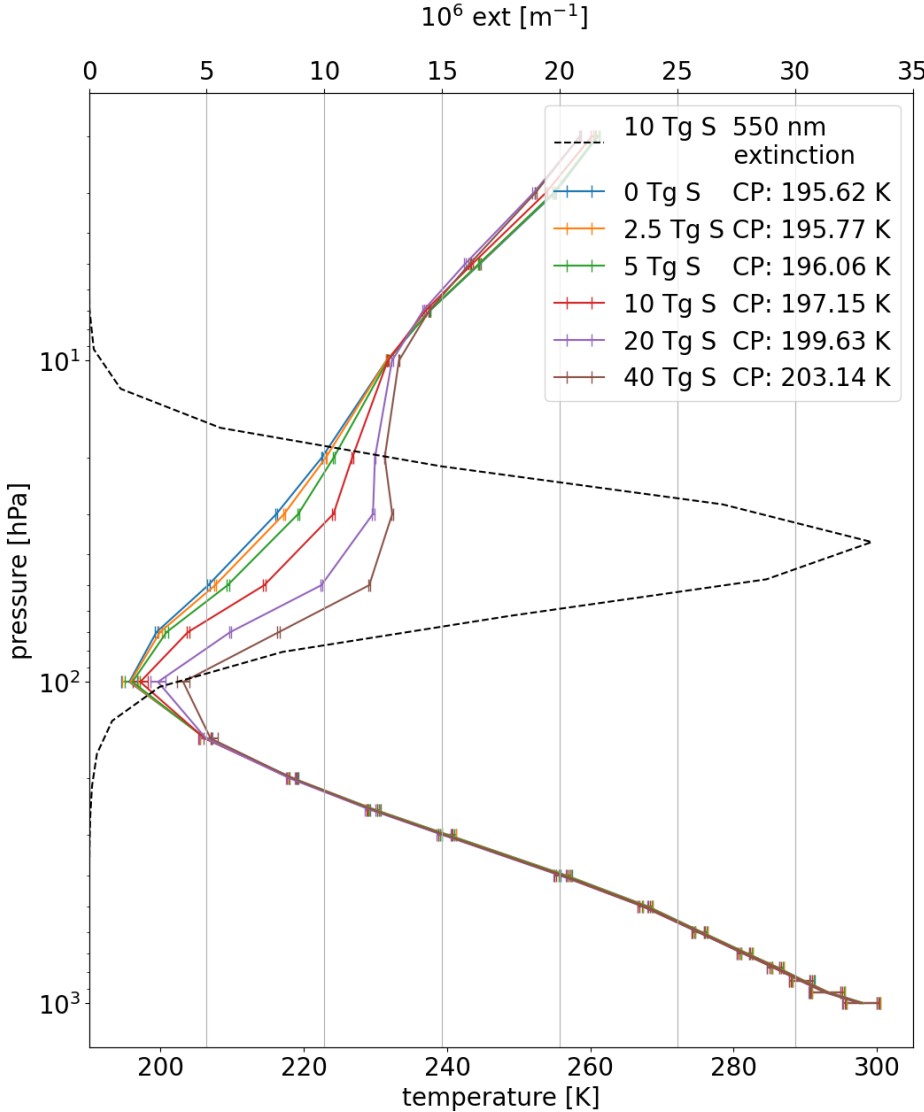

**Figure 3.** Inner tropical average of the temperature profiles for the five volcanically perturbed EVAens runs (2.5 Tg S, 5 Tg S, 10 Tg S, 20 Tg S and 40 Tg S) in September 1991 (three months after the eruption). The temperature of the respective cold point (CP) is indicated in the legend. The solid line represents the ensemble mean, the error bars symbolize the ensemble standard deviations. The 550 nm extinction values for the 10 Tg S aerosol profile are shown with a dashed black line.

Figure 4 shows changes in the temporal evolution of the surface temperature (a), cold point temperature (b), and 100 hPa specific humidity (c) with respect to the zero emission run in the inner tropics. Whereas the surface temperature decreases due

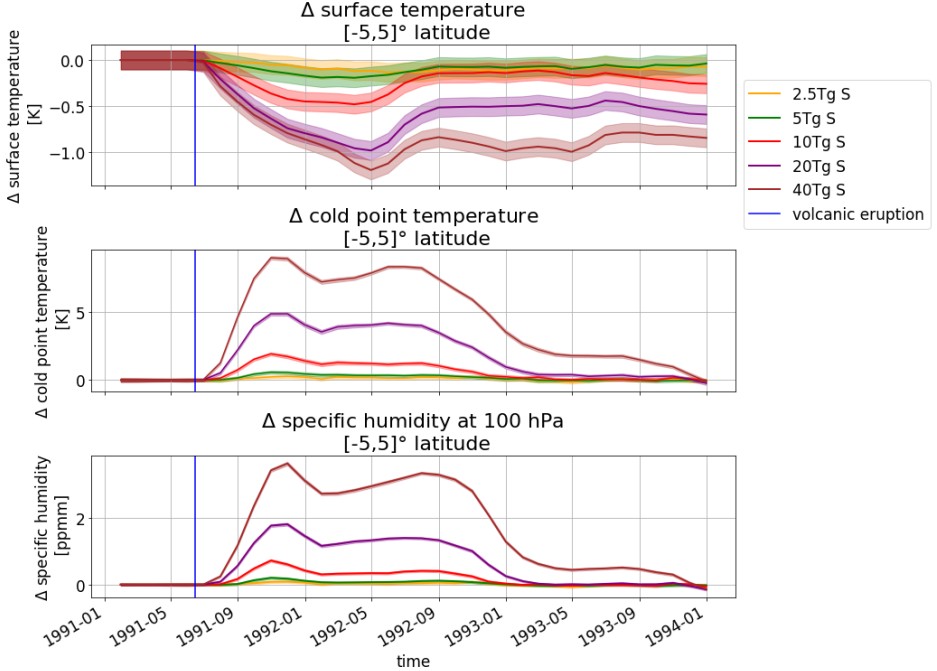

**Figure 4.** Temporal evolution of the inner tropical mean anomaly in surface temperature, cold point temperature, and stratospheric water vapour between volcanically perturbed and unperturbed ensemble runs. The ensemble means are shown with their standard errors. The time of the volcanic eruption is indicated by a vertical blue line.

to the scattering of incoming shortwave radiation by the volcanic aerosols, the cold point temperature increases due to warming of the tropopause layer by absorption of long wave radiation by the aerosol layer.

When considering the ensemble mean and its standard error even the temperature changes of the low emission runs (2.5 Tg S and 5 Tg S) are significantly different from the control run for short periods of time. With maximum values of 1.93 K the ensemble mean for the cold point temperature of the 10 Tg S is the lowest sulfur emission to reach a mean warming above the control run standard deviations[2], cold point values below this range could be found for single control run ensemble members due to internal variability. The higher emission scenarios (20 Tg S and 40 Tg S) have longer periods with cold point

temperature changes above the normally observed internal variability.

In addition the higher emission group shows a second amplification peak of the yearly cycle of the cold point temperatures between May and September 1992. In both the 20 Tg S and the 40 Tg S this second peak is reduced with respect to the first peak.

The monthly changes in cold point temperature are not linear either with respect to emitted sulfur mass or to AOD in the prominent IR waveband (compare Fig. A3, A4). Additionally the transient behaviour of the volcanic forcing - the build up

---

[2]The standard deviations are ten times larger than the standard error in case of our 100 member ensemble.





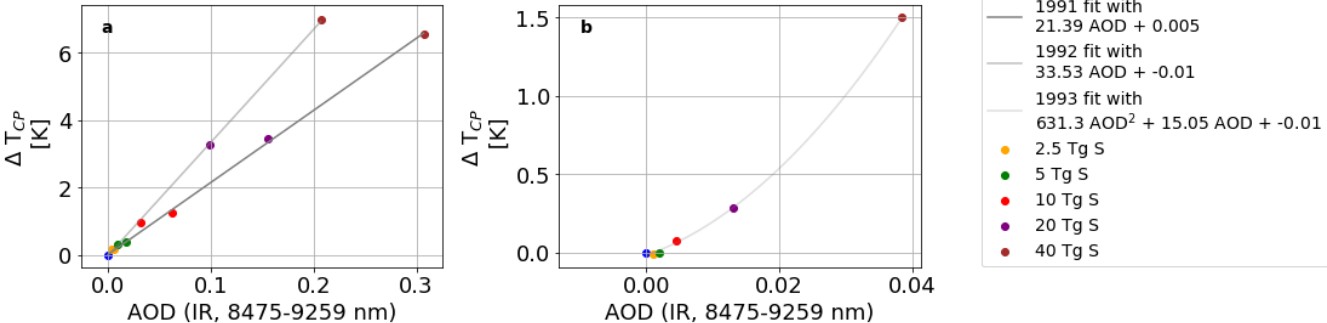

**Figure 5.** Yearly averages of adjusted tropical cold point temperature increases as a function of tropical AOD (IR, 8475-9259 nm) for the three examined years (1991-1993). A second order fit for each year with corresponding equation is shown.

phase in 1991, the approximately constant forcing in 1992, and the declining phase in 1993, which also differs between the individual eruption sizes - leads to a hysteretic behaviour. Nevertheless, when partitioning the cold point warmings into these individual phases, a linear relationship between IR-AOD and cold point warming arises for the years 1991 and 1992 (Fig. 5).

In 1993 this relationship becomes quadratic since the lower eruption strength are already ceasing to warm the TTL region.

The found cold point temperature increases are accompanied by an increase in the saturation water vapour pressure, reducing the "freeze trap" drying in the cold point region. The increased saturation water vapour pressure enables more water vapour to enter the lower stratosphere as shown in Fig. 4 (c) [3]. Consequently the evolution of the additional stratospheric water vapour

closely follows the evolution of the cold point temperature changes.

### 3.3 Effects on the tape recorder signal

The annual cycle of the tropical SWV is often described as the tape recorder signal (Mote et al., 1996): The variations of the tropical cold point temperatures controlling the water vapour entry into the stratosphere via the saturation water vapour pressure is imprinted on the stratospheric water vapour as music is imprinted on a tape. This leads to an annual cycle of bands

of high and low water vapour content propagating upwards in the stratosphere with the Brewer Dobson Circulation.

As the LW-heating by the volcanic aerosols will lead to increased tropical cold point temperatures, an increase of stratospheric water vapour is expected after volcanic eruptions with stratospheric aerosols. As shown in Sect. 3.2 the specific humidity shows an enhancement which does not stay constant and then declines but has an annual cycle like the tape recorder. In the following section we will investigate the seasonal cycle more closely.

Figure 6 and 7 show absolute SWV and the differences in the SWV content above 140 hPa with respect to the control run for all five eruption strengths. The maximum increases are found in the eruption year itself ranging from 0.1 ppmm for the 2.5 Tg S eruption to 3.5 ppmm for the 40 Tg S eruption. This corresponds to 5 %, respectively 160 %, of the unperturbed

---

[3]Here and in the following analysis we report the specific humidity as mass of water vapour per mass of moist air in ppmm values.





SWV values (see Fig. B1). The larger the eruption, the earlier the increase becomes visible and is significant. The additional SWV also follows the annual cycle of the tape recorder (Mote et al., 1996), showing maxima in the SWV enhancement around
September which then propagate upwards. This seasonal variation is also apparent in the behaviour of the tropopause and cold point heights: for scenarios with at least 10 Tg S onward the tropopause pressures and cold point pressures are higher in the northern hemispheric autumn; whereas in the 20 Tg S and 40 Tg S runs the volcanic forcing leads to higher pressure levels of the cold point from September 1991 until the end of 1992 accompanied by a seasonal signal in the SWV anomalies. Allowing for more water vapour to transit to the lower stratosphere.

In the 40 Tg S ensemble mean the second seasonal cycle is associated with an upward propagation of the SWV increases above 3 ppmm to even lower atmospheric pressures than in the preceding year. This behaviour can be attributed to the persistence of the high levels of aerosols in combination with the already enhanced SWV levels due to the presence of the volcanic aerosol in the previous year, the additional warming caused by the SWV and the lower lying cold point.

Shortly after the eruption a decrease in water vapour content above 50 hPa is visible. At these altitudes SWV increases with
height due to its production by methane oxidation[4]. Heating in the aerosol layer below leads to lofting of air parcels, bringing lower humidity air upwards to higher altitudes, where it causes a net reduction in humidity. However, this effect never exceeds 0.5 ppmm and is offset as soon as the lifted air becomes more moist due to enhanced SWV entry through the tropopause region.

---

[4]The MPI-ESM uses a parameterized methane oxidation scheme (Schmidt et al., 2013).





**Figure 6.** Tropical average in [-23,23]° latitude of WV above 140 hPa for sulfur injections of 2.5 Tg S, 5 Tg S, 10 Tg S, 20 Tg S and 40 Tg S as well as the PADS dataset. The WMO-tropopause pressure is indicated by a black line, the cold point pressure is shown as black dashed line. Absolute values are shown. In regions not covered by black crosses statistical significant difference between water vapour values of the perturbed and unperturbed runs (t-test at p=0.05) were found.





**Figure 7.** Tropical average in [-23,23]° latitude of WV anomalies above 140 hPa for the sulfur injections of 2.5 Tg S, 5 Tg S, 10 Tg S, 20 Tg S and 40 Tg S as well as the PADS dataset. The lowermost panel shows the MPI-GE historical simulations for Mt. Pinatubo using the PADS forcing data set discussed in Sect. 3.5. The WMO-tropopause pressure is indicated by a black line, the cold point pressure is shown as black dashed line. Differences with respect to the 0 Tg S control run are shown. In regions not covered by black crosses statistical significant difference between water vapour values of the perturbed and unperturbed runs (t-test at p=0.05) were found.





### 3.4 Intra-ensemble variability

The investigation of single ensemble members is of interest because - unlike for the ensemble mean - their physical characteristics would actually be "observable". Figure 8 shows seasonal averages of the specific humidity at the cold point as a function of cold point temperature for the inner tropical average in 1991 and 1992. The season SON was chosen as it is the period of highest SWV values in the lower stratosphere. Each point in Fig. 8 symbolizes one single ensemble member. Orange crosses show the theoretically possible maximum specific humidity curve calculated as the saturation humidity for water vapour over

ice at the average cold point temperature, using the corresponding exact solution for the Clausius Clapeyron equation by Murphy and Koop (2005)[5]. An approximation for the Clausius Clapeyron Equation at this temperature range with an 12 % increase of specific humidity per K is shown with a dashed grey line. This approximation is based on the assumption that the atmosphere around the cold point is saturated in the 0 Tg S run. The single ensemble members follow the slopes and values of the calculated water vapour amount at saturation in the inner tropics ([-5,5]° latitude). Next to water vapour, sublimated lofted

ice can also contribute to the SWV leading to specific humidity values exceeding the ones expected when calculating the the SWV based on the saturation water vapour alone as soon as temperatures rise again and the saturation water vapour increases. However at the location of the cold point this lofted ice would still be in the ice state and not accounted for in the specific humidity term.

In the inner tropics the found specific humidity values agree nicely with the values from the Clausius Clapeyron equation and

its approximated form of a 12 % per Kelvin. However this is only true for the inner tropics. For the entire tropics values up to around 1 ppmm lower could be found (compare Fig. C1). As the cold point temperatures increase with increasing latitude, higher specific humidity values would be expected at higher latitudes. However, the water vapour enters the tropics mainly in the inner tropical region and then spreads throughout the globe, leading to values lower than expected according to the Clausius Clapeyron equation.


In the first SON after the eruption in 1991 the SWV values and cold point temperatures are larger than in 1992. Additionally the separation by sulfur content is more pronounced in 1991. Table 1 lists the number of individual ensemble members lying within two standard deviations of the temperature or humidity value spread of the control run. In 1991 several individual group members up to the 5 Tg S run overlap with the control run spread as far as cold point temperature changes and humidity

values are concerned. The 10 Tg S run marks the emission strength at which single ensemble members start to be significantly different from the control run as only one ensemble member can not be distinguished from the control run using only the temperature values. The 20 Tg S run is the first emission strength with no control run overlap and for which all individual ensemble members show significant increase of SWV content and cold point temperatures in 1991 and even in 1992 where the lower emission runs start to show an increasing number of ensemble members overlapping with the control run values again.

This analysis shows the difficulty to register the SWV increases in observational data which collect data for a single realisa-

---

[5]Formula (7) giving the saturation water vapour pressure above ice $p_{ice}$ as $p_{ice} = exp(9.550426 - 5723.265/T + 3.53068 ln(T) - 0.00728332T)$ is used. This formula is valid for temperatures above 110 K. With the knowledge of $p_{ice}$ and the respective total pressure a calculation of the specific humidity is possible.





tions of a volcanic eruption. Although signals similar to those of individual eruptions can be produced by internal variability of an unperturbed scenario, our larger ensemble size allows to extract a robust signal which may be obscured in one single observational record.

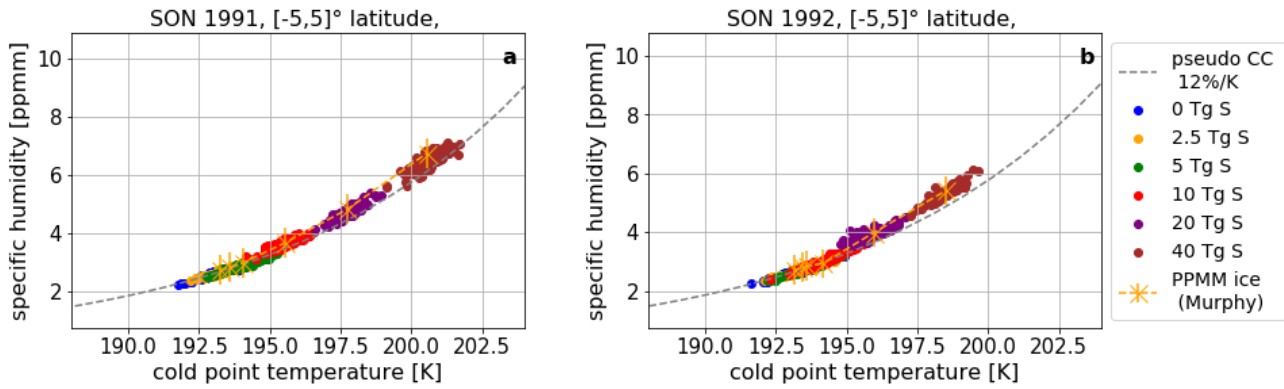

**Figure 8.** Seasonal averages of specific humidity at the cold point as a function of cold point temperature for SON 1991 (a) and 1992 (b). Values for each individual ensemble member are shown as dots for the inner tropics. An approximation (see text) for the Clausius Clapeyron equation at this temperature range with an 12 % increase of specific humidity per K is shown with a dashed grey line. The exact solution for the Clausius Clapeyron Equation over ice by Murphy and Koop (2005) is calculated for the average ensemble cold point temperatures and pressure and shown in orange.

**Table 1.** Number of ensemble members lying within the two standard deviations of the corresponding control run values with respect to temperature (T) or specific humidity (Q) or their union $(Q \cup T)$ in 1991 and 1992.

| Case | 2.5 Tg S | 5 Tg S | 10 Tg S | 20 Tg S | 40 Tg S |
|---|---|---|---|---|---|
| $Q_{1991}$ | 93 | 76 | 0 | 0 | 0 |
| $T_{1991}$ | 92 | 79 | 1 | 0 | 0 |
| $(Q \cup T)_{1991}$ | 93 | 82 | 1 | 0 | 0 |
| $Q_{1992}$ | 97 | 89 | 52 | 0 | 0 |
| $T_{1992}$ | 98 | 90 | 65 | 0 | 0 |
| $(Q \cup T)_{1992}$ | 99 | 90 | 65 | 0 | 0 |

**3.5 Comparison to the MPI-GE historical simulations (Mt. Pinatubo)**

The importance of the aerosol profile shape and particularly the extinction at the cold point for the SWV entry becomes apparent when comparing the results of the EVAens members to the Mt. Pinatubo period in the MPI-GE historical simulations.





In terms of the amount of emitted sulfur the EVAens simulations for 5 Tg S and 10 Tg S can be seen as bounds of recent estimates of the sulfur emission of the Mt. Pinatubo eruption of $(7.5 \pm 2.5)$ Tg S (Timmreck et al., 2018). As the PADS data

set describing the Mt. Pinatubo eruption is based on observational evidence rather than simulated as is the case for the EVA data sets only a range and not a set amount of emitted sulfur is given.

In Fig. 2, which shows the global mean TOA radiative imbalance for the MPI-GE historical simulations along with the EVAens simulations, the values for the Mt. Pinatubo eruption in the MPI-GE historical simulations lie between the 5 Tg S and the 10 Tg S imbalances in 1991. In 1992 the mean MPI-GE TOA imbalance is slightly more negative than the ensemble mean of even

the 10 Tg S run, but overall the deviations between the 10 Tg S run and Mt. Pinatubo run in the MPI-GE historical simulations are small compared to the standard error of the ensemble means.

However the tropical 1992 SWV anomalies in the historical simulations show a stronger increase than in the 10 Tg S EVAens simulations (absolute values are shown in Fig. 7, for percental changes see Fig. B1). In particular the seasonal cycle of the tape recorder is more strongly amplified in the Mt. Pinatubo run in 1992, where the SWV increase of 0.7 ppmm exceeds the

0.5 ppmm values of 1991. The heights the corresponding SWV increases reach also differ: the 0.5 ppmm signal propagates upwards to 20 hPa in the historical run, whereas in the 10 Tg S run the 0.5 ppmm signal only reaches pressure levels of up to 60 hPa.

The higher SWV values are caused by a stronger heating of the atmospheric region around the cold point controlling the indirect SWV entry. Fig. 9 shows cold point temperatures for the example of SON 1992. Mt. Pinatubo in the MPI-GE historical

simulations reaches values lying between those of the 10 Tg S and 20 Tg S run.



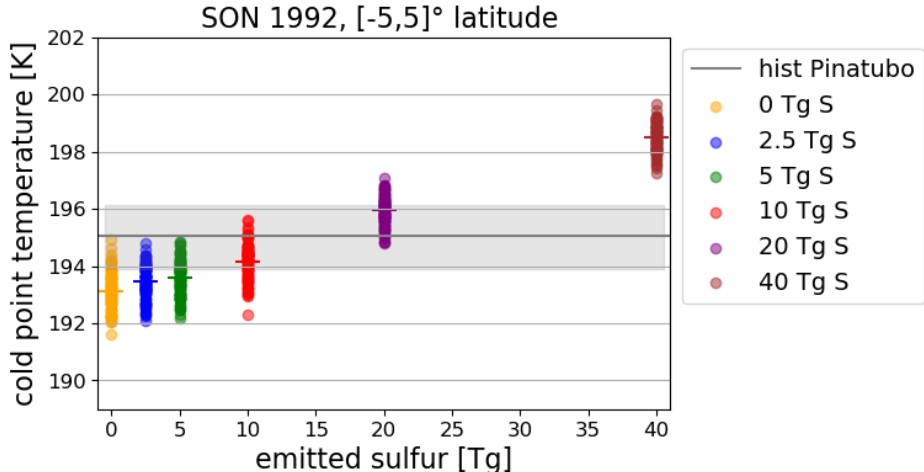

**Figure 9.** Average cold point temperature in the inner tropical region [-5,5]° latitude as a function of emitted sulfur for all EVAens members in
SON 1992. Each point symbolizes one ensemble member, the horizontal lines denote the ensemble mean. The average cold point temperature
range for the historical Mt Pinatubo eruptions (PADS) is shown as a grey line, the shaded region shows the extent between corresponding
maximum and minimum cold point temperature.

This can be understood when comparing the aerosol extinction profiles of EVA and PADS as proxy for the heating generated
by the aerosol layer (Fig. 10).

Although the PADS data set Mt. Pinatubo forcing has a peak value lying between the 5 Tg S and 10 Tg S EVA extinctions,
the extinction at the tropopause and cold point pressure is slightly higher than the 10 Tg S extinction in September 1991 (Fig.
10a and 10c). In September 1992 the PADS data set reaches values of the 20 Tg S EVA in the cold point region and a peak
extinction comparable to the 10 Tg S profile (Fig. 10b and 10d)[6]. As the warming of the cold point determines the SWV entry
anomaly, the extinction values at this point will have a significant impact.

In 1991 the 10 Tg S EVA extinction values at the cold point are only slightly weaker than the extinction values of the PADS
forcing set. The SWV values in the MPI-GE historical simulations are nearer to but not equal to those in the 10 Tg S as the
peak extinction values in the 10 Tg S are more than two times larger than the PADS extinction values and partially compensate
the lower values at the cold point. In 1992 the situation changes and the MPI-GE shows higher SWV values than the 10 Tg S
ensemble. The amplification of the SWV entry in the second SON after the eruption can be attributed to three phenomena. First,
the LW-heating was only building up in the northern hemisphere summer of 1991 and had not yet reached its maximum value
as can be seen in the plot for the cold point temperatures (Figure 4). Second, the peak extinction values of the PADS data set
start to exceed the 10 Tg S values in 1992, which may lead to higher SWV-entry values (Figure 11a). Third, the LW-extinction
at the cold point reaches a second, larger maximum around the northern summer of 1992 with values comparable to the 20

[6]A similar behaviour is apparent in the solar extinction bands (s. Appendix E1).



Tg S. This second maximum is not represented in the EVA-forcing (Figure 11b) and goes along with a decrease in the peak

forcing difference in the EVAens which lead to higher SWV entry values in the EVAens in 1991[7].

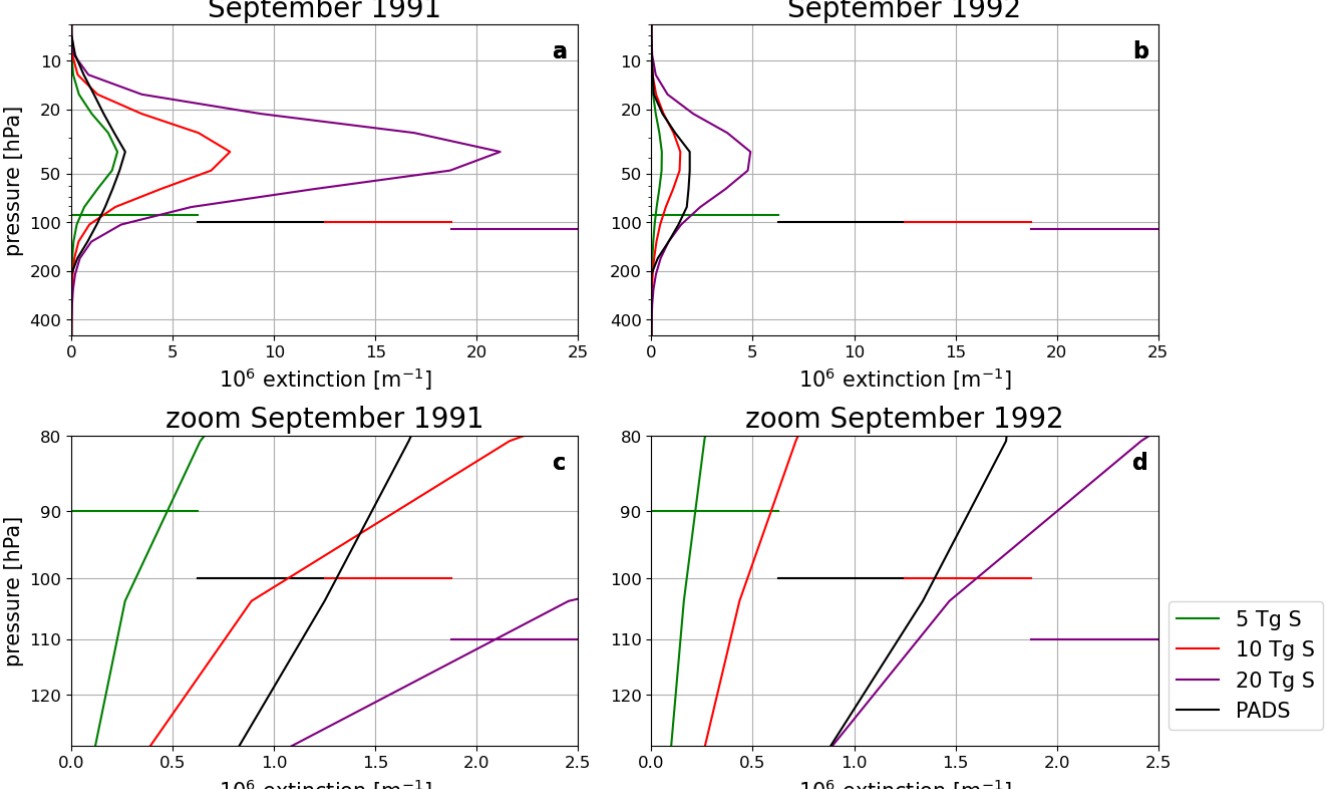

**Figure 10.** Average tropical aerosol extinction in the 8475-9259 nm infrared waveband from EVA for 5 Tg S, 10 Tg S, and 20 Tg S

emissions as well as from PADS in September 1991 and 1992. The horizontal lines indicate the pressure levels of the cold point in the region

between [-5,5]° latitude for each eruption strength. (a) and (b) show the entire profile. (c) and (d) are zooms on the cold point region.

---

[7]The larger difference between peak extinction and extinction at the cold point as well as the faster decline of extinction values compensated for the very

similar extinction values at the cold point in 1991.





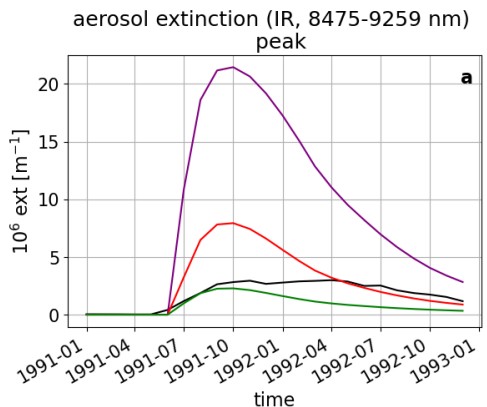
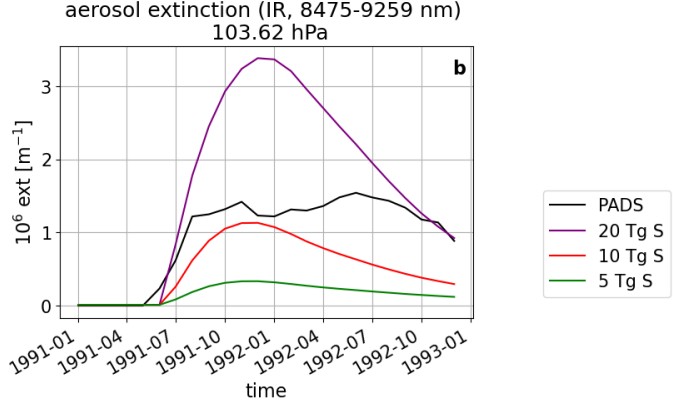

**Figure 11.** Temporal evolution of the aerosol extinction at the profile peak (a) and the cold point region (b) for the 8475-9259 nm IR-waveband.

### 3.6 Adjusted forcing caused by the SWV increases

In the following the adjusted SWV forcing is investigated using the one-dimensional model konrad, as described in Sect. 2.2. Fig. 12a shows flux anomalies between an unperturbed run and a run perturbed with the increased SWV levels of the 40 Tg S run but without the volcanic aerosols. Although SWV levels are increased within the complete stratospheric column, the main flux changes occur in the tropopause region where the enhancement is largest, the atmosphere denser and the effect of stratospheric water vapour on the radiative forcing strongest (Solomon et al., 2010). In this region, the incoming solar radia-

tion (SWd) is reduced by absorption, while at lower altitudes, there is no difference in the shortwave flux. This is caused by the SWV absorbing part of the solar spectrum which amongst others tropospheric water vapour completely absorbs at lower altitudes in the unperturbed run. The reflected solar radiation (SWu) at the surface is only changed negligibly since Δ SWd at the surface is also negligible. Consequently the SW contribution (SWd-SWu) to the adjusted forcing at the TOA is very small and negative - accounting for a slightly increased scattering due to the SWV.

The emitted long wave radiation (LWu) near the surface is unchanged. This is due to our setup: As the surface temperature is fixed in the perturbed run, the surface LWu is fixed as well. The SWV leads to an increase in the tropopause temperatures, following this increase the emitted long wave radiation at the main SWV levels is also increased as can be seen in the increased downward long wave radiation (LWd). The outgoing long wave radiation (LWu) above the tropopause region is substantially reduced as the SWV acts as a greenhouse gas and traps part of the outgoing radiation. This leads to the characteristic net

positive forcing of the greenhouse gas at the TOA.

A comparison to the instantaneous forcing (Fig. 12b) shows that the temperature adaptations in the stratosphere significantly amplify the SWV forcing. Whereas the atmospheric levels with increased SWV are warmed, stratospheric cooling is found in higher atmospheric levels, reducing the LWu. This effect has to build up though and consequently is more pronounced in the





adjusted SWV forcing.


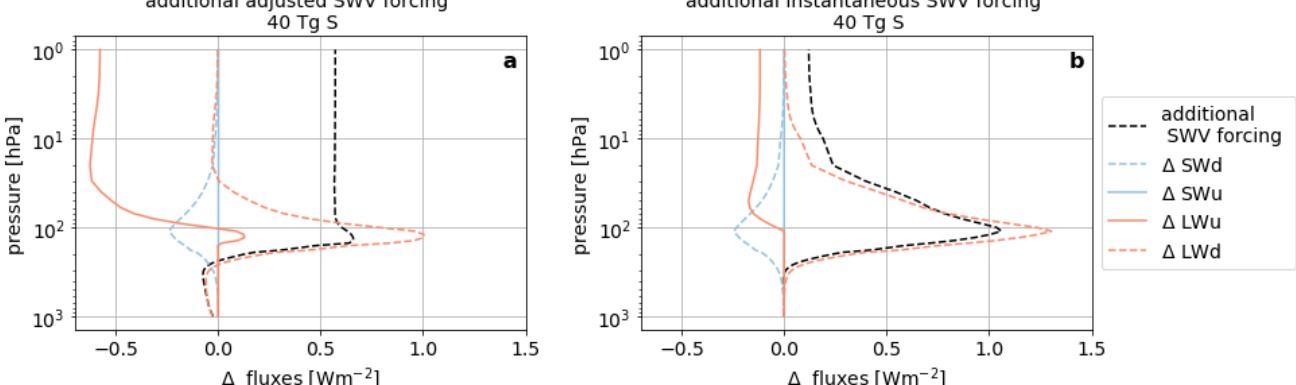

**Figure 12.** SW and LW contributions to the total adjusted SWV radiative forcing in the tropical stratosphere [-5,5]° latitude for November 1992 of the 40Tg S eruption. The upward fluxes are in solid lines and positive upward, the downward fluxes are in dashed lines and positive downward. The difference between the perturbed and unperturbed equilibrium state are shown. (a) adjusted forcing (b) instantaneous forcing.

The temporal evolution of the adjusted inner tropical SWV forcing is shown for all five eruptions strengths in Fig. 13. The adjusted forcing caused by the SWV, corresponding to the 2.5 Tg S to 10 Tg S run, is below the maximum fluctuations caused by internal variability denoted by the grey line in Fig. 13. The forcing found for these runs could be found for one ensemble member in the 0 Tg case as well, although the time span in which it would occur would most probably be shorter. The adjusted

SWV forcing for the 40 Tg S run reaches values of up to $0.65\,\mathrm{Wm^{-2}}$.

The signal evolution, especially in the 20 Tg S and 40 Tg S cases, matches the evolution already observed for the SWV increases in the tropical region, with two seasonal peaks. However, since also SWV at pressures lower than 100 hPa contribute to the total forcing - although to a smaller extent - and the transport out of the tropical region with the BDC is slow the first peak forcing times are longer than the peak specific humidity values at 100 hPa in Fig. 4.

When considering clouds the stratospherically adjusted forcing caused by the additional SWV is slightly increased by 0.1 $\mathrm{Wm^{-2}}$ at most (compare Appendix Fig. E2). As the clouds reflect part of the down coming SW radiation some of the SW bands, which could be completely absorbed while traveling through the complete stratosphere and troposphere, are not absorbed entirely. The additional SWV in the stratosphere increases the absorption in these bands and reduces the outgoing SW radiation. This leads to an increase of the forcing.




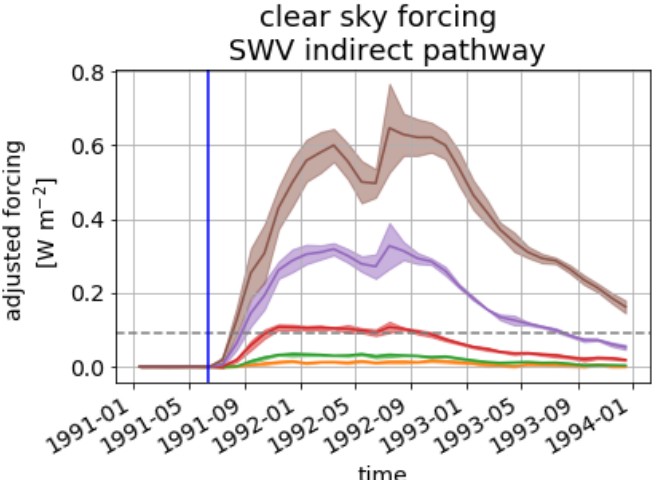

**Figure 13.** Time evolution of the TOA adjusted clear sky forcing in the tropical region [-5,5]° latitude for the ensemble mean SWV increases caused by all eruption strengths. The blue line marks the eruptions time, the dashed grey line shows the threshold up to which deviations in radiative fluxes can be caused by internal variability. Shaded areas indicate the flux anomaly range originating from the standard deviations of the SWV profiles are plotted to visualize the signal range.

The monthly SWV forcing does not exhibit a linear behaviour with respect to the mass of emitted sulfur or main IR AOD waveband (Fig. A5, A6) as was to be expected since the monthly cold point temperatures also did not change in a linear man-ner with respect to the emitted sulfur mass and the temperature increases showed a hysteretic behaviour. However again, when averaging out the seasonal dependencies and partitioning the time after the eruption in the signal build up phase (1991, after

the eruption), the phase of approximately constant forcing (1992), and the phase of declining signal (1993) the relationship is quasi linear[8]. Deviations from the linear trend are introduced by the higher eruption strength not reaching the maximum forcing values as fast as the lower eruption strengths in 1991. Additionally, the lower eruption strengths already are relaxing to the background state in 1993. Compared to the cold point - AOD relation the signal build up and relaxation back to the ground state is damped as the transport of the SWV into the stratosphere and out of the tropical region takes place over longer

timescales than the cold point warming or cooling respectively.

---

[8]Although the relationship is of second order the second order term does not lead to a large contribution within the examined region.





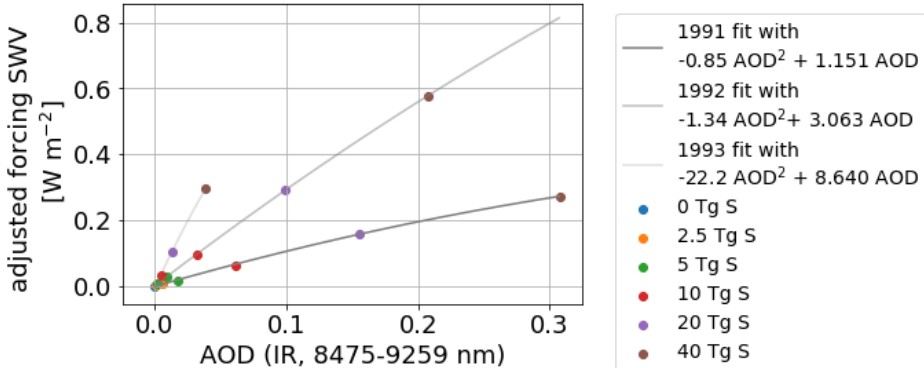

**Figure 14.** Yearly averages of adjusted forcing due to the additional SWV as a function of AOD (IR, 8475-9259 nm) for the three examined years (1991-1993). A second order fit for each year with corresponding equation is shown.

As the SWV forcing counteracts the volcanic forcing its relation to the aerosol forcing is of interest. Fig. 15a shows the tropical aerosol forcing as calculated using the double radiation call in the MPI-ESM. For the evaluation of the forcing in the double radiation call different conventions exists as far as the evaluation at the TOA or the tropopause are concerned (e.g. compare Forster et al. (2016)). As the double radiation call is used to determine an instantaneous forcing the readout at the tropopause level would be following the standard convention as defined by Hansen et al. (2005) or in the IPCC. However we present both values for the entire and inner tropics to allow for an easy comparison to other studies. The relative magnitude of the SWV adjusted forcing with respect to the aerosol forcing increases approximately linearly until September 1992 and then reaches a constant values of around 2.5 % for the readout at the tropopause and 4 % percent for the readout at the TOA (Fig. 15b).





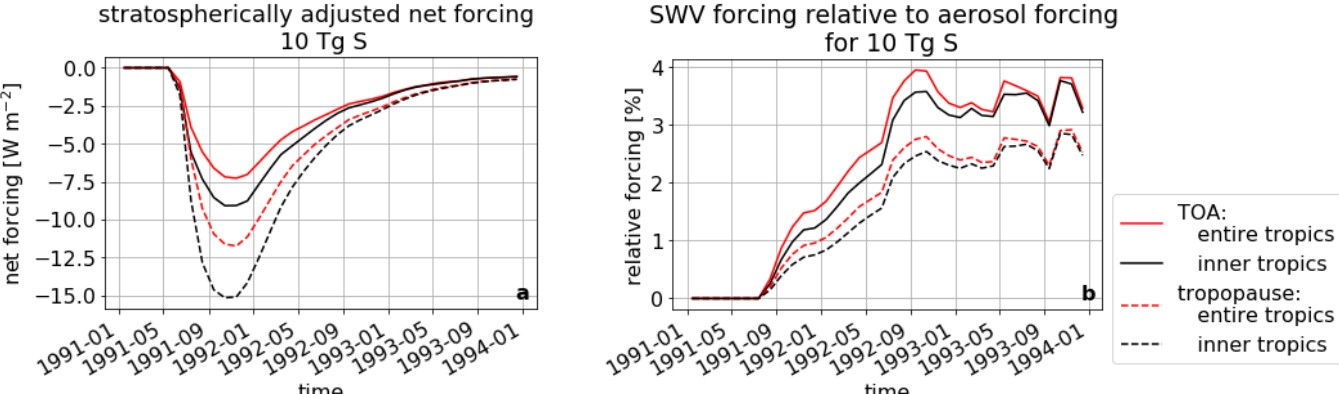

**Figure 15.** (a) Time evolution of the tropical aerosol forcing for the 10 Tg S run as calculated using the double radiation call in the MPI-ESM. The forcing is evaluated for the inner and entire tropics at the tropopause and the TOA. (b) Time evolution of the percentage of aerosol forcing counterbalanced by the SWV forcing in the inner and entire tropics .

## 4 Discussion

### 4.1 Magnitude of SWV increases due to indirect mechanism

In general, the annual cycle of the tropical tropopause temperatures account for a variation of SWV content of $\pm$ 1.4 ppmv $\approx$ 0.87 ppmm around the mean background at 110 hPa in SAGE II data of the early 1990s (Mote et al., 1996). In the MPI-GE the variations of the SWV tape recorder signal at the same height is lower: it reaches $\pm$ 0.69 ppmm at most. However the SAGE II data which Mote et al. (1996) used fell in the era of the Mt. Pinatubo eruption and may be biased due to an amplification of the seasonal signal by the presence of the aerosol layer, an effect leading to maximum deviations of up to $\pm$ 1.0 ppmm in the 10 Tg S scenarios of the EVAens simulations.

In the case of volcanic perturbations, the alterations in SWV content caused by the indirect entry mechanism via tropopause warming after volcanic eruptions can surpass the SWV variations due to the annual cycle. In our simulations, deviations of more than $\pm$ 0.69 ppmm are produced in the simulations for emissions equal to or larger than 10 Tg S eruption, which is an upper bound of the emission estimate for the Mt. Pinatubo eruption (Timmreck et al., 2018). The SWV anomaly caused by the 2.5 Tg S eruption is comparable to changes caused by the Quasi Biennial Oscillation (QBO) which are [0.1-0.2] ppmm according to the regression analysis by Dessler et al. (2013), whereas the 10 Tg S has an impact stronger than the changes in the Brewer Dobson Circulations (maximum 0.4 ppmm).

In climate change studies the stratospheric water vapour is also affected by doubled $CO_2$ concentrations. The SST increase leads to a consequent atmospheric humidity increase amounting up to 6-10 % in the stratosphere, with values exceeding 10 % in the lower stratosphere (Wang et al. (2020)). These changes of SWV due to increased $CO_2$ levels are comparable to increases caused by the smaller eruptions of 2.5 Tg S and 5 Tg S. However, the SWV changes due to volcanic eruptions are





only temporary.

## 4.2   Comparison to studies based on reanalysis data

While our model study has the advantage that the large ensemble size allows to determine the statistical significance of SWV
increases and the spread of responses to a volcanic eruption due to internal variability of the earth system, a model study
is always limited by the ability of the model to represent the earth system realistically. Individual models may differ in the
parametrization of convection, the entry of water vapour into the stratosphere and tropopause height. The aerosol profiles used
in this study are artificial in case of the EVAens and will include uncertainties of the retrieval in case of the PADS data set. As
our results are therefore at first only representative for the model used in our study, a comparison to results based on or derived
from observational evidence of SWV changes after volcanic eruptions is desirable.


During the Mt. Pinatubo period observations of SWV are scarce, especially as solar occultation measurements done by
HALOE and SAGE II in this time period suffered from aerosol interference. Therefore, the data usage is discouraged (e.g.
Fueglistaler et al. (2013)). Although Fueglistaler et al. (2013) mention "anomalously large anomalies" of SWV values shortly
after the Mt. Pinatubo eruption in SAGE II data, they also warn that the measurements may be biased by aerosol artifacts since
the SWV signal is not present in the HALOE data. We will therefore compare our results not directly to satellite observations
but to the outcome of two regression analysis studies of SWV entry fed by reanalysis input by Dessler et al. (2014) and Tao
et al. (2019).

Dessler et al. (2014) studied the different contributors to SWV entry at 82 hPa slightly above the tropical tropopause in their
model by using a regression analysis on data output from a trajectory model fed by MERRA reanalysis input (Rienecker et al.,
2011). They found a maximum increase of 0.34 ppmv $\approx$ 0.21 ppmm in the residual which partially overlapped with the AOD
increase caused by the Mt. Pinatubo eruption. This value is lower than our finding of up to (0.5-0.7) ppmm increase in the first
and second post eruption year. Our higher value may be explained by the different quantification approaches: whereas Dessler
et al. (2014) indirectly quantified the SWV increase due to Mt. Pinatubo in the residuals after subtracting contributing terms
like the Brewer Dobson Circulation (BDC), we directly quantified the additional SWV entry by taking the difference between
the MPI-GE historical ensemble with volcanic aerosol and the EVAens control run. This allowed us to avoid the subtraction
of SWV entry caused by the volcanic eruption but attributed to other sources. For example, in the Dessler et al. (2014) study,
the BDC term is described by the heating in the 82 hPa region, but part of this may be caused by the presence of the volcanic
aerosol layer. Our method also eliminates other sources of SWV, that may have caused the SWV increase prior to the Mt.
Pinatubo eruption in the residual of Dessler et al. (2014). Additionally the tropopause in our historical simulation is located at
pressures larger than 100 hPa. Dessler et al. (2014) report that 82 hPa is slightly above the tropopause. Their possibly higher
lying tropopause may also explain part of the difference.



Tao et al. (2019) use MERRA-2 (Gelaro et al., 2017), JRA-55 (Kobayashi et al., 2015) and ERAinterim (Dee et al., 2011)

reanalysis data for the Mt. Pinatubo eruption in their trajectory model. The SWV increase attributed to the volcanic eruption

ranges between $0.4$ ppmv $\approx 0.2$ ppmm (ERAinterim) and $0.8$ ppmv $\approx 0.5$ ppmm (MERRA-2 and JRA-55). Only MERRA-2

explicitly accounts for the volcanic aerosol (Fujiwara et al. (2017)); the corresponding $0.5$ ppmm SWV increase is in good

agreement with the SWV increases found in the first post eruption year in our historical simulations, although our SWV increase

in the second post eruption year of $0.7$ ppmm is larger. As analyzed in Sect. 3.5 this higher increase is mainly caused by the

height of the aerosol layer with respect to the cold point. Since the aerosol profiles are imported on fixed pressure levels, with-

out prescribing the distance between the tropopause and the aerosol layer, and the tropopause heights differ amongst various

models, the comparison between different studies and models would be facilitated if not only the total volcanic forcing but also

the heating in the region were reported when quantifying and comparing volcanically induced SWV increases. A tropopause

located at larger pressure in the second post eruption summer in the reanalysis data could explain the differences between the

maximum values of $0.5$ ppmm by Tao et al. (2019) using the MERRA-2 reanalysis and our results amounting up to $0.7$ ppmm

in the second post eruption summer as it would lead to a lower heating rate in the cold point region and consequently a reduced

water vapour entry compared to the control years. Additionally our model does not include interactive chemistry - $H_2O$ sinks

could reduce the amount of water vapour and especially the build up in the second poster eruption summer. However Löffler

et al. (2016) also found a stronger SWV increase in the second post eruption summer when investigating the perturbations of

stratospheric water vapour using nudged chemistry-climate model simulations with prescribed aerosol for the Mt. Pinatubo

eruption. The SWV increases for the Mt. Pinatubo eruption reach values of up to 35 % compared to the unperturbed run in

the inner tropical average. Thus our finding of an increase of 25 % above the unperturbed levels for the historical simulations

(compare Fig. B1) lies between the estimates from reanalysis data by Tao et al. (2019) and the chemistry climate studies by

Löffler et al. (2016)

In the EVAens, where the temporal evolution of the extinction profile does not lead to as drastic changes in the vertical shape

as in the PADS forcing data set, the temporal evolution of the increase in SWV is similar to the evolution found by Tao et al.

(2019), with only one prominent maximum. Maximum values of SWV increase are in the range of $0.2$ ppmm and $0.7$ ppmm

for the 5 Tg S and 10 Tg S EVAens runs - covering the estimated sulfur emission range of Mt. Pinatubo and in agreement with

Tao et al. (2019).


## 4.3 SWV contributions due to the indirect and direct injection

The main difference between the direct and indirect mechanism is that the indirect pathway is active for the entire life time

of the volcanic aerosol in the lower stratosphere, whereas the direct injection is a singular event. There is no study known to

us comparing the SWV entry due to both events within one framework. Although up to 80 % of the eruption volume can be

water vapour (Coffey (1996)), rapid condensation can remove 80-90 % of this humidity on the way to the stratosphere (Glaze

et al. (1997)). There are only a few cases of direct injections reported, which cover relatively small eruption events. The water

vapour within the eruption column which reached the stratosphere did not lead to elevated SWV above background for more



than a week locally in the few cases reported, i.e the eruption of Kasatochi (2008) with maximum SWV content of 9 ppmv $\approx$ 5.6 ppmm (Schwartz et al., 2013) lasting for around 1 day, the eruption of Calbuco (2015) with 10 ppmv $\approx$ 6.2 ppmm values

for approximately a week (Sioris et al. (2016a)) and the eruption of Mt. Saint Helens with $(64 \pm 4)$ ppmv $\approx (40 \pm 2)$ ppmm (Murcray et al. (1981)) detectable above background for around a week, but only on local and not global scale. In the case of the Mt. Pinatubo eruption the model estimates for the direct injection tended to have a larger range and maximum values than the estimates based on observational data (Joshi and Jones, 2009). In contrast the indirect pathway allows for slower, but more continuous, raised stratospheric water vapour levels ultimately spread throughout the globe but with lower peak values. These

enhanced SWV levels are detectable in our ensemble mean even years after the actual volcanic eruption if the emitted sulfur amount is large enough (larger than 10 Tg S). Depending on the explosivity of the volcano the relative contributions of the direct and indirect injection mechanism to SWV increases should change: for small eruptions the direct injection will lead to a relatively high increase in SWV, which is short lived and spatially confined. For the larger eruptions this short lived SWV enhancement is followed by a relatively stable increase of SWV which spreads throughout the entire globe, dominating the

SWV increase due to the volcanic eruption.

## 4.4 SWV Forcing

In our simulations the adjusted forcing caused by the increase of SWV in the tropical region amounts to maximally 2.5 to 4 percent of the tropical aerosol forcing in the same time frame for the 10 Tg S eruption. However, although a decline of the tropical forcing within the three year time-frame is found, the forcing around the complete globe will last longer than the

impact of the volcanic aerosols. The decline in tropical stratospheric water vapour is caused by its transport to the poles by the Brewer Dobson Circulation, leading to a regional shift of the location of the SWV forcing (see Fig. F1). Forster and Shine (2002) found the polar SWV forcing to be 2.5 times stronger than the tropical forcing: Since in the polar region the tropopause height is lower and the water vapour content is generally lower, SWV increases of the same magnitude have a much larger impact there than in the tropical region. Based on the factor of 2.5, the 40 Tg S run 1993 values would cause an adjusted

forcing of up to 0.5 $\mathrm{Wm}^{-2}$ in the polar region, whereas the aerosol forcing is back to the background levels of the 2.5 Tg S run by 1993. This shift in relative magnitude is one of the factors contributing to the positive TOA imbalance at the end of 1993 (for 20 Tg S and 40 Tg S in Fig. 2).

For the eruption of Mt. Pinatubo with $(7.5 \pm 2.5)$ Tg S emitted, earlier estimates of SWV forcing exist. Joshi and Shine

(2003) calculated the global SWV forcing to be 0.1 $\mathrm{Wm}^{-2}$. Our 10 Tg S adjusted forcing results of up to 0.11 $\mathrm{Wm}^{-2}$ are nearer to the estimate by Joshi and Shine (2003) than our 5 Tg S results of up to 0.03 $\mathrm{Wm}^{-2}$. In 1992, the adjusted forcing in the inner tropics lies between [0.02 - 0.03] $\mathrm{Wm}^{-2}$ for the 5 Tg S and [0.06 - 0.11] $\mathrm{Wm}^{-2}$ for the 10 Tg S scenarios. The better agreement of our 10 Tg S adjusted SWV forcing with Joshi and Shine (2003) value for the Mt. Pinatubo SWV forcing can be attributed to two points: First, the form and location of the forcing profile with respect to the cold point is crucial when

comparing model results of increased SWV levels and the consequent changes in SWV forcing (s. Sect. 3.5). This may contribute to differences between values from Joshi and Shine (2003) and our analysis. Second, the polar forcing will be stronger



than the tropical forcing which was calculated with konrad. Using the ratio of polar forcing to tropical forcing by Forster and Shine (2002) the polar estimate would be [0.05 - 0.08] $\mathrm{Wm}^{-2}$ for 5 Tg S and [0.15 - 0.28] $\mathrm{Wm}^{-2}$ for 10 Tg S.

500       In their study on direct SWV entry for the Krakatau eruption Joshi and Jones (2009) found a LW-forcing of $+(0.33 \pm 0.09)$ $\mathrm{Wm}^{-2}$ for a direct entry above 100 hPa of 1.5 ppmv $\approx 0.933$ ppmm using the downward TOA heat flux, climate feedback parameter and near surface temperature changes. They chose a relatively high estimate of SWV increase, which would approximately correspond to the SWV increases in our 20 Tg S eruption run. As the TOA-SW contribution to our forcing is negligible (s. Fig. 12) a comparison to our total forcing is possible. The corresponding value of [0.21 - 0.33] $\mathrm{Wm}^{-2}$ is in the lower part
of the range found by Joshi and Jones (2009), which is likely caused mainly by the restriction of our study to the tropical region.

      Krishnamohan et al. (2019) also mentioned the contribution of aerosol induced SWV changes to the flux changes at the TOA in a geoengineering study, however they did not explicitly calculate it. The amount of short wave forcing they attribute to additional SWV is much higher than our value and also positive. Presumably these differences are caused by the forcing
including tropospheric adjustments and using a 2 x $CO_2$ reference frame and thus can not be compared to our study.

      In order to put the adjusted radiative SWV forcing due to the indirect pathway into a broader context we compare it with SWV forcing due to anthropogenic $CO_2$ and methane releases: the rate of radiative forcing increase due to $CO_2$ in the 2000s[9] reached values of almost 0.03 $\mathrm{Wm}^{-2}\mathrm{year}^{-1}$ and a total of $(1.82 \pm 0.19)\,\mathrm{Wm}^{-2}$ for the time frame from 1750 to 2011, whereas
the additional radiative forcing due to methane in the same time frame is $(0.48 \pm 0.05)\,\mathrm{Wm}^{-2}$ (Myhre et al., 2013)[10]. The forcing caused by the SWV entering the stratosphere via the indirect entry mechanism exceeds the yearly increase of forcing due to $CO_2$ in the 2000s starting with the 5 Tg S run. The peak adjusted tropical SWV forcing for the 40 Tg S scenario amounts to one third of the total $CO_2$ forcing (due to the accumulated emissions from 1750 to 2011) and is larger than the forcing due to methane emissions for the same period.

**4.5   Predictability of responses**

The increase in cold point temperature and SWV are delayed events. Time is required for it to build up the signals as the aerosols warm the cold point region allowing more water vapour transit into the stratosphere. An approximately stable phase with fluctuations due to the seasonal cycle is attained a couple of months after eruption. The cold point warming has stabilized. The SWV forcing is mainly determined by the additional SWV entering the stratosphere each month as the tropopause region
is the region in which WV has the strongest radiative effect. The corresponding build up of tropical SWV forcing due to an accumulation of SWV in the stratosphere is therefore counteracted by the transport to higher altitudes and from the tropics to the polar region by the Brewer Dobson Circulation.

Due to the transient character of the processes - the build up of the forcing and the consequent decline as well as the time

---

[9]Atmospheric $CO_2$ concentration increased from $(278 \pm 2)$ ppmm to $(390.5 \pm 0.2)$ ppmm.
[10]Methane concentration increased from $(722 \pm 25)$ ppb to $(1803 \pm 2)$ ppb.

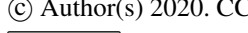



shift between effect and response - our analysis showed that a linear relationship between AOD and cold point warming, or
SWV, can not be determined for the entire era after the volcanic eruption. Time phases have to be considered as well to account
for the resulting hysteresis. Other parameters additionally constrain the cold point warming and SWV response. The season
during which the eruption occurs will influence the impact of the aerosols: The CP warming will be most effective during the
peak times of the tape recorder signal around September. Each volcanic eruption is different and parameters like the eruption
location, amount of emitted sulfur, and specifically the location of the aerosol layer will have a massive influence both on cold
point warming and SWV entry/forcing. Feedbacks like the SWV forcing in the TTL will enhance the CP warming and lead to
higher SWV forcing at same AODs eventually before the aerosols fall out.

Nevertheless an approximately linear relationship[11] can be found in our simulations for CP warming and SWV forcing with
respect to IR-AOD after the volcanic eruptions when taking yearly averages. These results however must be interpreted with
care and the derived formulas are only applicable for tropical eruptions occurring in June and reaching to similar heights in the
stratosphere.

Generally, the knowledge both of the respective CP warming in the inner tropics for a specific volcanic eruption and the
respective background SWV in the cold point region allows for a relatively accurate prediction of the increase in SWV levels
when using a 12 % SWV increase per Kelvin warming in the mean CP values.

## 5   Conclusion and Outlook

Our study of EVAens and the historical simulations led us to draw the following conclusions:

1. The analysis of the ensemble runs showed the difficulty to extract the volcanic signal in the SWV if only individual
   observations are available as internal variability of SWV in the control run could produce SWV values as large as the
   variation found for some of the 10 Tg S ensemble members in our simulations. Ensemble mean SWV increases are
   already significant[12] in the 2.5 Tg S eruption.

2. The increase in stratospheric water vapour does not remain constant throughout year but can vary with the seasons. An
   amplification of the seasonal cycle could be observed, especially in the 20 Tg S and 40 Tg S scenarios.

3. The average WV at the cold point entering the stratosphere after volcanic eruptions can be approximated with only minor
   errors using the mean saturation water vapour pressure over ice at the respective average tropical cold point temperature
   for all investigated aerosol profiles of the EVAens. When given a base value of an unperturbed atmospheric state, a 12 %
SWV increase per K increase in cold point temperature is a good first estimate for eruption strengths up to 40 Tg S.

4. The comparison of the idealized EVAens simulations and MPI-GE historical simulations of Mt. Pinatubo show that
   neither the simulated TOA radiative imbalance nor the estimated amount of emitted stratospheric sulfur suffice to con-

---

[11]The second order term does not contribute significantly since all input values are smaller than 1 in the AOD range of the volcanic eruptions.

[12]T-test comparison to reference run with p=0.05.





strain the SWV increases. However the aerosol layer shape and height with respect to the tropopause play a crucial and dominating role when estimating the SWV increases.

5. The adjusted tropical forcing caused by the additional SWV last longer than the forcing caused by the aerosol layer itself and its contribution to the total volcanic forcing grows with time as the aerosols fall out. For the 20 Tg and the 40 Tg scenarios the TOA radiative imbalance shows a positive value by the end of the simulation. Part of this positive TOA radiative imbalance can be attributed to the forcing by the additional SWV.

     6. When considering also the time dependence an approximately linear relationship between yearly averaged tropical cold
point warming/adjusted SWV forcing and IR-AOD can be deduced. This relationship however only holds for comparable eruptions occurring at the same time within the year in the tropics. Additionally the final eruption height and aerosol profile shape has to match the one used within our framework.

Based on our study different follow up questions for future investigations arise. Our study only focuses on the indirect injection and although other studies focus on the direct injection no study know to us combines both effects allowing for
a direct comparison within one single framework. Using integrated plume models for a combined study would allow the quantification of the entire SWV changes and for an estimation of their relative importance. As we use no interactive chemistry our estimate of the $H_2O$ might be overestimating the SWV from the indirect injection remaining in the stratosphere. A study using interactive chemistry would enable the assessment of the impact of $H_2O$ sinks on the volcanically induced SWV increase as well as ozone chemistry. Of particular interest are the impact of the additional $H_2O$ on the oxidation process of $SO_2$, as well
as on sulfate particle formation and growth, and a follow up study on this topic could enable a lower boundary estimate to be made for the SWV increases and the changed aerosol lifetime (compare Case et al. (2015), Kilian et al. (2020)). Additionally the mechanism of the indirect injection has implications beyond that of a volcanic eruption: As geoengineering scenarios also apply sulfur derivatives in the stratosphere, an investigation of the long term SWV signal within these scenarios may be of interest as well.

*Code and data availability.* Details to the EVAens primary data is published with the first publication on the EVAens by Azoulay et al. (2020). The MPI-ESM historical simulations are described by Maher et al. (2019). The 1D RCE model konrad is available online under https://github.com/atmtools/konrad.





**Appendix A: Scaling of different physical parameters altered by the presence of volcanic aerosols**

Following the discussion in Sect. 3.1, 3.2 and 3.6 the graphs for TOA imbalance (Fig. A1), surface temperature (Fig. A2 ), cold

point temperature changes (Fig. A3) and SWV forcing (Figure A5 ) are shown which each ensemble mean divided by the mass

of emitted sulfur. In case of the cold point temperature changes and the SWV forcing the dependence of monthly mean values

of cold point temperature change and SWV forcing are shown as a function of AOD in the IR waveband of [8475,9259] nm in

Fig. A4 and A6.

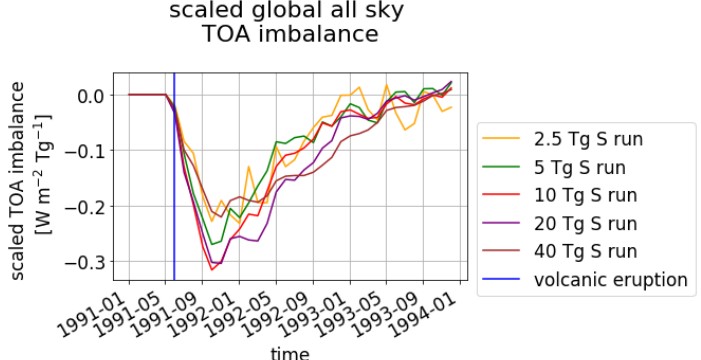

**Figure A1.** Scaled top of the atmosphere (TOA) imbalance for the five volcanically perturbed EVAens runs (2.5 Tg S, 5 Tg S, 10 Tg S, 20
Tg S and 40 Tg S). The ensemble means are shown. The vertical blue line marks the eruption time. All TOA imbalances are divided by the
mass of emitted sulfur. Incoming fluxes are defined positive, outgoing fluxes are defined negative.

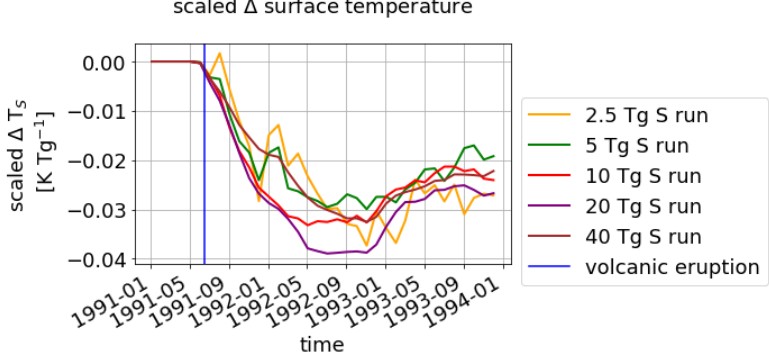

**Figure A2.** Scaled surface temperature for the five volcanically perturbed EVAens runs (2.5 Tg S, 5 Tg S, 10 Tg S, 20 Tg S and 40 Tg S).
The ensemble means are shown. The vertical blue line marks the eruption time. All surface temperatures are divided by the mass of emitted
sulfur.





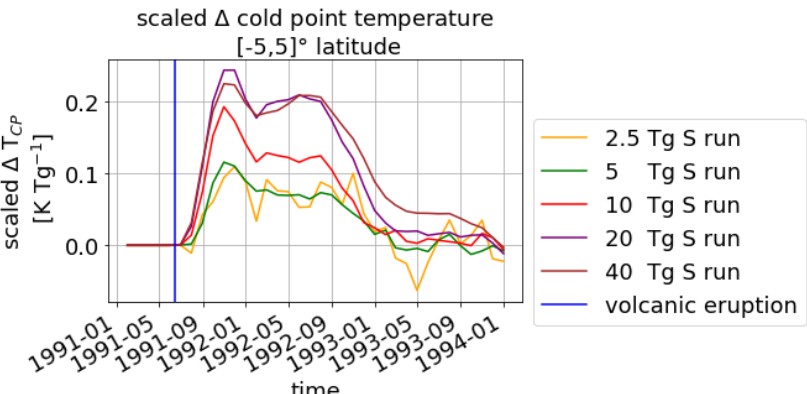

**Figure A3.** Scaled temporal evolution of the mean CP temperature anomaly. The time of the volcanic eruption is indicated by a vertical blue line. All changes in cold point temperature are divided by the mass of emitted sulfur.

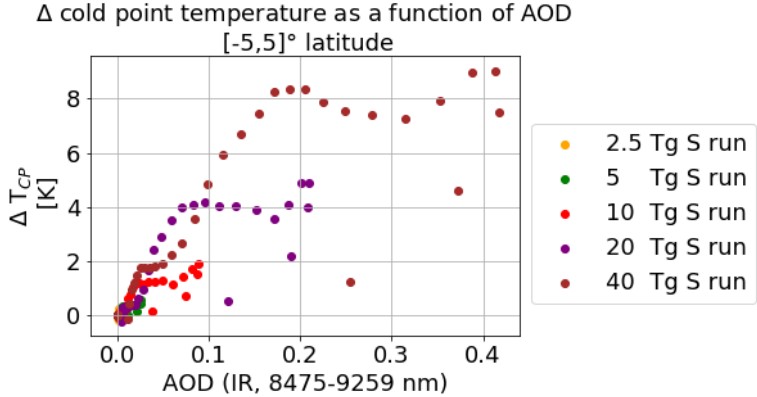

**Figure A4.** Scatter plot of AOD in the IR waveband (8475-9250 nm) and CP temperature anomaly for all time steps in the first 2.5 years after the eruption.





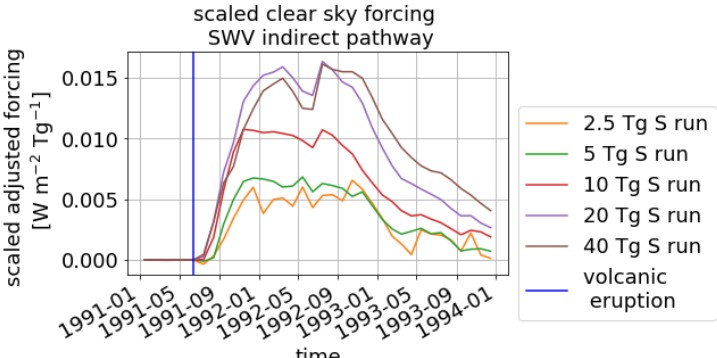

**Figure A5.** Scaled time evolution of the adjusted clear sky SW-forcing in the tropical region [-5,5]° latitude for all eruption strengths. The blue line marks the eruptions time. All clear sky forcings are divided by the mass of emitted sulfur. Incoming fluxes are defined positive, outgoing fluxes are defined negative

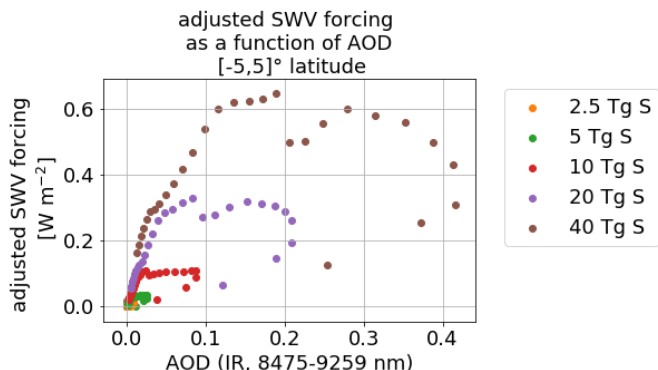

**Figure A6.** Scatter plot of AOD in the IR waveband (8475-9250 nm) and the clear sky SWV forcing for all time steps in the first 2.5 years after the eruption.

## Appendix B: Percental changes in the tape recorder signal

Complementary to the plot in Sect. 3.3 we show the differences in water vapour between perturbed and unperturbed state with respect to the unperturbed state in percent.







**Figure B1.** Percental difference in water vapour above 140 hPa in the tropical average over [-23,23]° latitude for the pure sulfur injections of the EVAens (2.5 Tg S, 5 Tg S, 10 Tg S, 20 Tg S and 40 Tg S). The lowermost panel shows the MPI-GE historical simulations for Mt. Pinatubo using the PADS forcing data set. The height of the WMO-tropopause is indicated by a black line, the cold point pressure is shown as black dashed line. In regions not covered by black crosses statistical significant difference between stratospheric water vapour values of the perturbed and unperturbed runs (t-test at p=0.05) were found.





## Appendix C: Intra-ensemble variability in the entire tropics

Here we show the complementary plots to those in Sect. 3.4 for the intra-ensemble variability in the entire tropics [-23,23]° latitude.

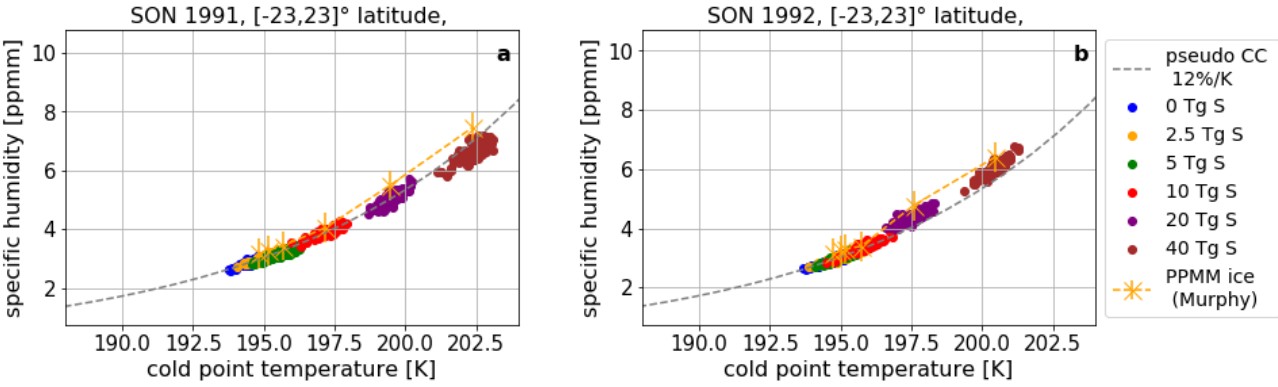

**Figure C1.** Seasonal averages of specific humidity at cold point as a function of cold point temperature for SON 1991 (a) and 1992 (b) accounting for the entire tropics. Values for each individual ensemble member are shown as dots for the entire tropics. An approximation (see text) for the Clausius Clapeyron equation at this temperature range with an 12 % increase of specific humidity per K is shown with a dashed grey line. The exact solution for the Clausius Clapeyron Equation over ice by Murphy and Koop (2005) is calculated for the average ensemble cold point temperatures and pressure and shown in orange.

## Appendix D: aerosol extinction profiles - 550 nm solar waveband

Complementary to the discussion on the infrared extinction profiles in Sect. 3.5 the 550 nm solar waveband is shown in the following plots.

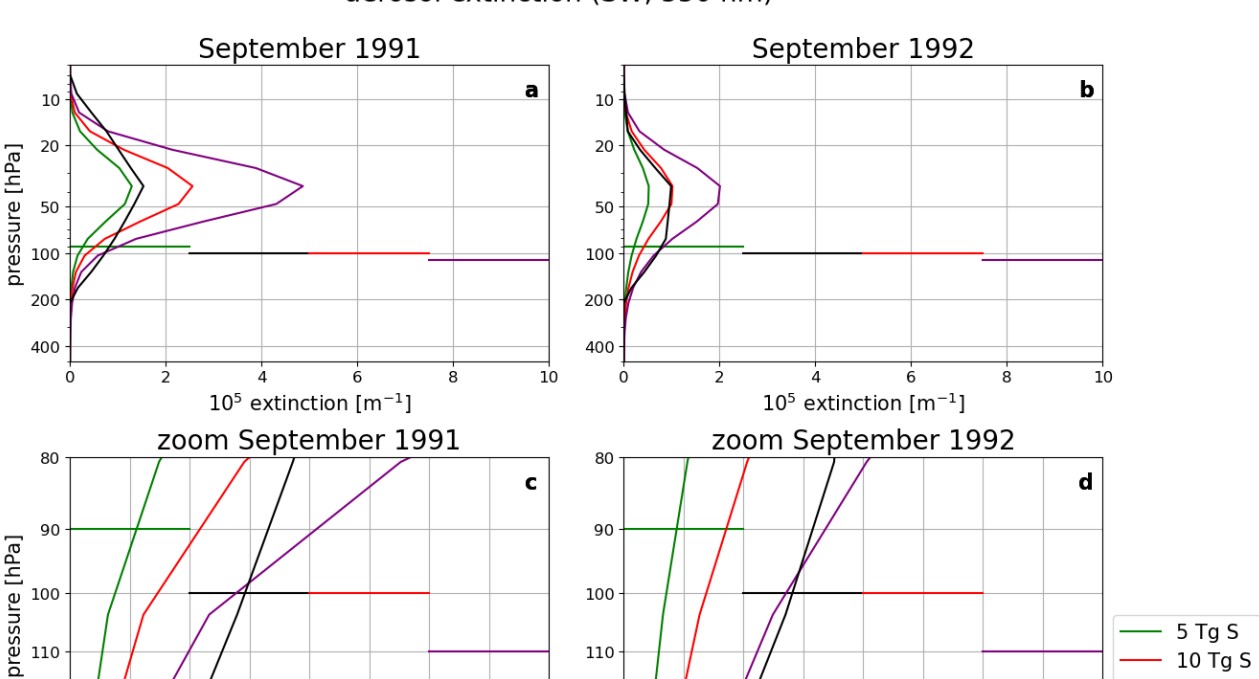

**Figure D1.** Tropical average of the aerosol extinction profile in the 550 nm solar waveband for the EVA forcing corresponding to 5 Tg S, 10 Tg S and 20 Tg S as well as the PADS forcing for the Mt. Pinatubo eruption.

**Appendix E: SWV forcing - SW component**

Fig. E1 shows the very small contribution of the SW component to the total adjusted SWV forcing presented in Sect. 3.6.





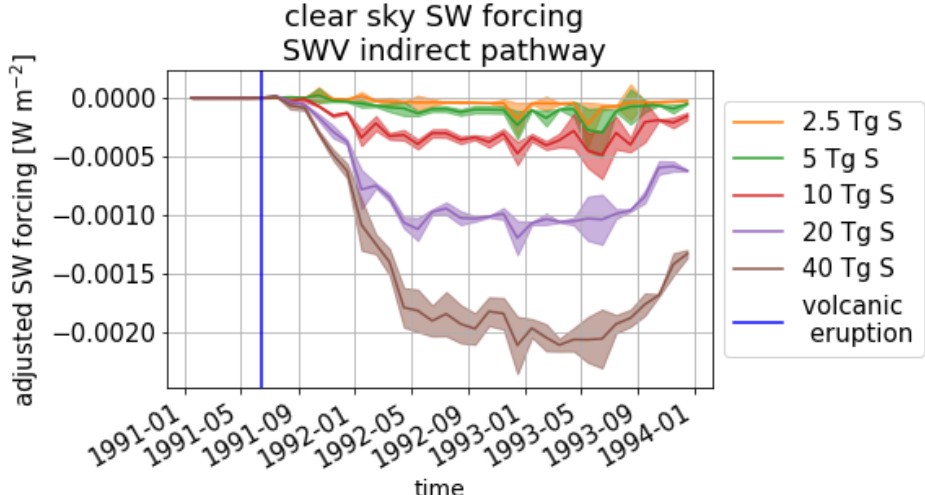

**Figure E1.** Time evolution of the SW-component to the adjusted clear sky forcing in the tropical region [-5,5]° latitude for the ensemble mean SWV increases caused by all eruption strengths. The blue line marks the eruptions time. The flux range originating from the standard deviations of the SWV profiles are plotted to visualize the signal range.

The total forcing and its SW component for the cloudy sky case as discussed in Sect. 3.6 is shown in Fig. E2.

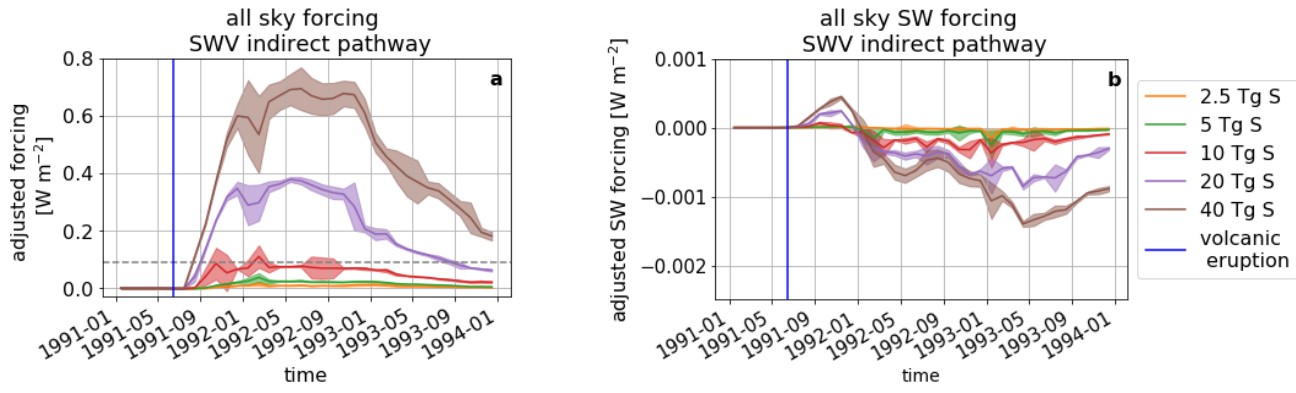

**Figure E2.** Time evolution of the (a) total (b) SW-component to the adjusted all sky forcing in the tropical region [-5,5]° latitude for the ensemble mean SWV increases caused by all eruption strengths. The blue line marks the eruptions time. The flux range originating from the standard deviations of the SWV profiles are plotted to visualize the signal range.

## Appendix F: Global spread of the stratospheric water vapour

Figure F1 shows the spread of the additional SWV around the globe as mentioned in the Discussion Sect. 4.4.





**Figure F1.** Difference in stratospheric water vapour content at 70 hPa as a function of time and latitude. Black crosses mark the regions of statistical significance of the data (Mann Whitney U Test at p=0.05).





*Author contributions.* CK, CT and HS designed the study. CK conducted the analysis/investigation and wrote the paper. SD contributed to the development of the methodology to calculate the SWV forcing with konrad and the interpretation of the corresponding result. AA set up
the EVAens simulations. CT, SD and HS contributed to the writing of the paper.

*Competing interests.* The authors declare that they have no conflict of interest.

*Acknowledgements.* This research has been supported by the Deutsche Forschungsgemeinschaft (DFG) Research Unit VolImpact (FOR2820) within the projects VolDyn (TO 967/2-1) and VolClim (TI 344/2-1). SD and CK were/are members of the International Max Planck Research School (IMPRS). The data was processed using CDO (https://code.mpimet.mpg.de/projects/cdo/embedded/cdo.pdf) and using the computing
facilities at the Deutsche Klimarechenzentrum (DKRZ). We thank Lukas Kluft for advising us on the usage of the 1D RCE model konrad.



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

Contributions of Volcanic Aerosols to Decadal Changes in the Stratospheric Circulation, Geophysical Research Letters, 44, 10,780–
10,791, https://doi.org/10.1002/2017GL074662, 2017.

Driscoll, S., Bozzo, A., Gray, L. J., Robock, A., and Stenchikov, G.: Coupled Model Intercomparison Project 5 (CMIP5) simulations of cli-
mate following volcanic eruptions, Journal of Geophysical Research: Atmospheres, 117, n/a–n/a, https://doi.org/10.1029/2012JD017607,
2012.

Forster, P. M., Richardson, T., Maycock, A. C., Christopher J. Smith, C. J., Samset, B. H., Myhre, G., Timothy Andrews, Timothy Pincus,
R., and Schulz, M.: Recommendations for diagnosing effective radiative forcing from climate models for CMIP6, Journal of Geophysical
Resarch:$_{A}tmosphere, 121, 12 460 - -12 475, https : //doi.org/10.1002/2016JD025320, 2016.$



Forster, P. M. d. F. and Shine, K. P.: Assessing the climate impact of trends in stratospheric water vapor, Geophysical Research Letters, 29, 10–1–10–4, https://doi.org/10.1029/2001GL013909, 2002.

Fueglistaler, S., Dessler, A. E., Dunkerton, T. J., Folkins, I., Fu, Q., and Mote, P. W.: Tropical tropopause layer, Reviews of Geophysics, 47, 1606, https://doi.org/10.1029/2008RG000267, 2009.

Fueglistaler, S., Liu, Y. S., Flannaghan, T. J., Haynes, P. H., Dee, D. P., Read, W. J., Remsberg, E. E., Thomason, L. W., Hurst, D. F., Lanzante,
J. R., and Bernath, P. F.: The relation between atmospheric humidity and temperature trends for stratospheric water, Journal of Geophysical Research: Atmospheres, 118, 1052–1074, https://doi.org/10.1002/jgrd.50157, 2013.

Fujiwara, M., Wright, J. S., Manney, G. L., Gray, L. J., Anstey, J., Birner, T., Davis, S., Gerber, E. P., Harvey, V. L., Hegglin, M. I., Homeyer, C. R., Knox, J. A., Krüger, K., Lambert, A., Long, C. S., Martineau, P., Molod, A., Monge-Sanz, B. M., Santee, M. L., Tegtmeier, S., Chabrillat, S., Tan, D. G. H., Jackson, D. R., Polavarapu, S., Compo, G. P., Dragani, R., Ebisuzaki, W., Harada, Y., Kobayashi, C., McCarty,
W., Onogi, K., Pawson, S., Simmons, A., Wargan, K., Whitaker, J. S., and Zou, C.-Z.: Introduction to the SPARC Reanalysis Intercomparison Project (S-RIP) and overview of the reanalysis systems, Atmospheric Chemistry and Physics, 17, 1417–1452, https://doi.org/10.5194/acp-17-1417-2017, 2017.

Gelaro, R., McCarty, W., Suárez, M. J., Todling, R., Molod, A., Takacs, L., Randles, C., Darmenov, A., Bosilovich, M. G., Reichle, R., Wargan, K., Coy, L., Cullather, R., Draper, C., Akella, S., Buchard, V., Conaty, A., da Silva, A., Gu, W., Kim, G.-K., Koster, R., Lucchesi, R.,
Merkova, D., Nielsen, J. E., Partyka, G., Pawson, S., Putman, W., Rienecker, M., Schubert, S. D., Sienkiewicz, M., and Zhao, B.: The Modern-Era Retrospective Analysis for Research and Applications, Version 2 (MERRA-2), Journal of Climate, Volume 30, 5419–5454, https://doi.org/10.1175/JCLI-D-16-0758.1, 2017.

Giorgetta, M. A., Jungclaus, J., Reick, C. H., Legutke, S., Bader, J., Böttinger, M., Brovkin, V., Crueger, T., Esch, M., Fieg, K., Glushak, K., Gayler, V., Haak, H., Hollweg, H.-D., Ilyina, T., Kinne, S., Kornblueh, L., Matei, D., Mauritsen, T., Mikolajewicz, U., Mueller, W., Notz, D.,
Pithan, F., Raddatz, T., Rast, S., Redler, R., Roeckner, E., Schmidt, H., Schnur, R., Segschneider, J., Six, K. D., Stockhause, M., Timmreck, C., Wegner, J., Widmann, H., Wieners, K.-H., Claussen, M., Marotzke, J., and Stevens, B.: Climate and carbon cycle changes from 1850 to 2100 in MPI-ESM simulations for the Coupled Model Intercomparison Project phase 5, Journal of Advances in Modeling Earth Systems, 5, 572–597, https://doi.org/10.1002/jame.20038, 2013.

Glaze, L. S., Baloga, S. M., and Wilson, L.: Transport of atmospheric water vapor by volcanic eruption columns, Journal of Geophysical
Research: Atmospheres, 102, 6099–6108, https://doi.org/10.1029/96JD03125, 1997.

Hall, T. M. and Waugh, D.: Tracer transport in the tropical stratosphere due to vertical diffusion and horizontal mixing, Geophysical Research Letters, 24, 1383–1386, https://doi.org/10.1029/97GL01289, 1997.

Hansen, J., Sato, M., Ruedy, R., Nazarenko, L., Lacis, A., Schmidt, G. A., Russell, G., Aleinov, I., Bauer, M., Bauer, S., Bell, N., Cairns, B., Canuto, V., Chandler, M., Cheng, Y., Del Genio, A., Faluvegi, G., Fleming, E., Friend, A., Hall, T., Jackman, C., Kelley, M., Kiang, N.,
Koch, D., Lean, J., Lerner, J., Lo, K., Menon, S., Miller, R., Minnis, P., Novakov, T., Oinas, V., Perlwitz, J., Perlwitz, J., Rind, D., Romanou, A., Shindell, D., Stone, P., Sun, S., Tausnev, N., Thresher, D., Wielicki, B., Wong, T., Yao, M., and Zhang, S.: Efficacy of climate forcings, Journal of Geophysical Research: Atmospheres, 110, https://doi.org/10.1029/2005JD005776, https://agupubs.onlinelibrary.wiley.com/doi/abs/10.1029/2005JD005776, 2005.

Ilyina, T., Six, K. D., Segschneider, J., Maier-Reimer, E., Li, H., and Núñez-Riboni, I.: Global ocean biogeochemistry model HAMOCC:
Model architecture and performance as component of the MPI-Earth system model in different CMIP5 experimental realizations, Journal of Advances in Modeling Earth Systems, 5, 287–315, https://doi.org/10.1029/2012MS000178, 2013.



Joshi, M. M. and Jones, G. S.: The climatic effects of the direct injection of water vapour into the stratosphere by large volcanic eruptions, Atmospheric Chemistry and Physics, 9, 6109–6118, https://doi.org/10.5194/acp-9-6109-2009, 2009.

Joshi, M. M. and Shine, K. P.: A GCM Study of Volcanic Eruptions as a Cause of Increased Stratospheric Water Vapor, Journal of Climate, 16, 3525–3534, https://doi.org/10.1175/1520-0442(2003)016<3525:AGSOVE>2.0.CO;2, 2003.

Jungclaus, J. H., Fischer, N., Haak, H., Lohmann, K., Marotzke, J., Matei, D., Mikolajewicz, U., Notz, D., and von Storch, J. S.: Characteristics of the ocean simulations in the Max Planck Institute Ocean Model (MPIOM) the ocean component of the MPI-Earth system model, Journal of Advances in Modeling Earth Systems, 5, 422–446, https://doi.org/10.1002/jame.20023, 2013.

Kilian, M., Brinkop, S., and Jöckel, P.: Impact of the eruption of Mt Pinatubo on the chemical composition of the stratosphere, Atmospheric Chemistry and Physics, 20, 11 697–11 715, https://doi.org/10.5194/acp-20-11697-2020, https://acp.copernicus.org/articles/20/11697/2020/, 2020.

Kluft, L., Dacie, S., Buehler, S. A., Schmidt, H., and Stevens, B.: Re-Examining the First Climate Models: Climate Sensitivity of a Modern Radiative–Convective Equilibrium Model, Journal of Climate, 32, 8111–8125, https://doi.org/10.1175/JCLI-D-18-0774.1, 2019.

Kobayashi, S., Ota, Y., Harada, Y., Ebita, A., Miyaoka, M., Onoda, H., Onogi, K., Kamahori, H., Kobayashi, C., Endo, H., Miyaoka, K., and Takahshini, K.: The JRA-55 Reanalysis: General Specifications and Basic Characteristics, Journal of the Meteorological Society of Japan. Ser. II, 93, 5–48, https://doi.org/10.2151/jmsj.2015-001, 2015.

Krishnamohan, K.-P. S.-P., Bala, G., Cao, L., Duan, L., and Caldeira, K.: Climate system response to stratospheric sulfate aerosols: sensitivity to altitude of aerosol layer, Earth System Dynamics, 10, 885–900, https://doi.org/10.5194/esd-10-885-2019, 2019.

Löffler, M., Brinkop, S., and Jöckel, P.: Impact of major volcanic eruptions on stratospheric water vapour, Atmospheric Chemistry and Physics, 16, 6547–6562, https://doi.org/10.5194/acp-16-6547-2016, 2016.

Maher, N., Milinski, S., Suarez-Gutierrez, L., Botzet, M., Dobrynin, M., Kornblueh, L., Kröger, J., Takano, Y., Ghosh, R., Hedemann, C., Li, C., Li, H., Manzini, E., Notz, D., Putrasahan, D., Boysen, L., Claussen, M., Ilyina, T., Olonscheck, D., Raddatz, T., Stevens, B., and Marotzke, J.: The Max Planck Institute Grand Ensemble: Enabling the Exploration of Climate System Variability, Journal of Advances in Modeling Earth Systems, 11, 2050–2069, https://doi.org/10.1029/2019MS001639, 2019.

Marsland, S. J., Haak, H., Jungclaus, J. H., Latif, M., and Röske, F.: The Max-Planck-Institute global ocean/sea ice model with orthogonal curvilinear coordinates, Ocean Modelling, 5, 91–127, https://doi.org/10.1016/S1463-5003(02)00015-X, 2003.

Mauritsen, T., Bader, J., Becker, T., Behrens, J., Bittner, M., Brokopf, R., Brovkin, V., Claussen, M., Crueger, T., Esch, M., Fast, I., Fiedler, S., Fläschner, D., Gayler, V., Giorgetta, M., Goll, D. S., Haak, H., Hagemann, S., Hedemann, C., Hohenegger, C., Ilyina, T., Jahns, T., Jimenéz-de-la Cuesta, D., Jungclaus, J., Kleinen, T., Kloster, S., Kracher, D., Kinne, S., Kleberg, D., Lasslop, G., Kornblueh, L., Marotzke, J., Matei, D., Meraner, K., Mikolajewicz, U., Modali, K., Möbis, B., Müller, W. A., Nabel, J. E. M. S., Nam, C. C. W., Notz, D., Nyawira, S.-S., Paulsen, H., Peters, K., Pincus, R., Pohlmann, H., Pongratz, J., Popp, M., Raddatz, T. J., Rast, S., Redler, R., Reick, C. H., Rohrschneider, T., Schemann, V., Schmidt, H., Schnur, R., Schulzweida, U., Six, K. D., Stein, L., Stemmler, I., Stevens, B., Storch, J.-S., Tian, F., Voigt, A., Vrese, P., Wieners, K.-H., Wilkenskjeld, S., Winkler, A., and Roeckner, E.: Developments in the MPI–M Earth System Model version 1.2 (MPI–ESM1.2) and Its Response to Increasing CO 2, Journal of Advances in Modeling Earth Systems, 11, 998–1038, https://doi.org/10.1029/2018MS001400, 2019.

Mote, P. W., Rosenlof, K. H., McIntyre, M. E., Carr, E. S., Gille, J. C., Holton, J. R., Kinnersley, J. S., Pumphrey, H. C., Russell, J. M., and Waters, J. W.: An atmospheric tape recorder: The imprint of tropical tropopause temperatures on stratospheric water vapor, Journal of Geophysical Research: Atmospheres, 101, 3989–4006, https://doi.org/10.1029/95JD03422, 1996.



Murcray, D. G., Murcray, F. J., Barker, D. B., and Mastenbrook, H. J.: Changes in stratospheric water vapor associated with the mount st. Helens

eruption, Science (New York, N.Y.), 211, 823–824, https://doi.org/10.1126/science.211.4484.823, 1981.

Murphy, D. M. and Koop, T.: Review of the vapour pressures of ice and supercooled water for atmospheric applications, Quarterly Journal of the Royal Meteorological Society, 131, 1539–1565, https://doi.org/10.1256/qj.04.94, 2005.

Myhre, Shindell, Bréon, Collins, Fuglestvedt, Huang, Koch, Lamarque, Lee, Mendoza, Nakajima, Robock, Stephens, and Takemura and Zhang: Anthropogenic and Natural Radiative Forcing: Climate Change 2013: The Physical Science Basis., Contribution of Working Group I to the

Fiith Assessment Report of the Intergovernmental Panel of Climate Change, https://www.ipcc.ch/site/assets/uploads/2018/02/WG1AR5_Chapter08_FINAL.pdf, 2013.

Reick, C. H., Raddatz, T., Brovkin, V., and Gayler, V.: Representation of natural and anthropogenic land cover change in MPI-ESM, Journal of Advances in Modeling Earth Systems, 5, 459–482, https://doi.org/10.1002/jame.20022, 2013.

Rienecker, M. M., Suarez, M. J., Gelaro, R., Todling, R., Bacmeister, J., Liu, E., Bosilovich, M. G., Schubert, S. D., Takacs, L., Kim, G.-K.,

Bloom, S., Chen, J., Collins, D., Conaty, A., da Silva, A., Gu, W., Joiner, J., Koster, R. D., Lucchesi, R., Molod, A., Owens, T., Pawson, S., Pegion, P., Redder, C. R., Reichle, R., Robertson, F. R., Ruddick, A. G., Sienkiewicz, M., and Woollen, J.: MERRA: NASA's Modern-Era Retrospective Analysis for Research and Applications, Journal of Climate, 24, 3624–3648, https://doi.org/10.1175/JCLI-D-11-00015.1, 2011.

Robock, A.: Volcanic eruptions and climate, Reviews of Geophysics, 38, 191–219, https://doi.org/10.1029/1998RG000054, 2000.

Robrecht, S., Vogel, B., Grooß, J.-U., Rosenlof, K., Thornberry, T., Rollins, A., Krämer, M., Christensen, L., and Müller, R.: Mechanism of ozone loss under enhanced water vapour conditions in the mid-latitude lower stratosphere in summer, Atmospheric Chemistry and Physics, 19, 5805–5833, https://doi.org/10.5194/acp-19-5805-2019, 2019.

Rosenlof, K. H.: Changes in water vapor and aerosols and their relation to stratospheric ozone, Comptes Rendus Geoscience, 350, 376–383, https://doi.org/10.1016/j.crte.2018.06.014, 2018.

Rosenlof, K. H., Oltmans, S. J., Kley, D., Russell, J. M., Chiou, E.-W., Chu, W. P., Johnson, D. G., Kelly, K. K., Michelsen, H. A., Nedoluha, G. E., Remsberg, E. E., Toon, G. C., and McCormick, M. P.: Stratospheric water vapor increases over the past half-century, Geophysical Research Letters, 28, 1195–1198, https://doi.org/10.1029/2000GL012502, 2001.

Schmidt, H., Rast, S., Bunzel, F., Esch, M., Giorgetta, M., Kinne, S., Krismer, T., Stenchikov, G., Timmreck, C., Tomassini, L., and Walz, M.: Response of the middle atmosphere to anthropogenic and natural forcings in the CMIP5 simulations with the Max Planck Institute Earth

system model, Journal of Advances in Modeling Earth Systems, 5, 98–116, https://doi.org/10.1002/jame.20014, 2013.

Schneck, R., Reick, C. H., and Raddatz, T.: Land contribution to natural CO 2 variability on time scales of centuries, Journal of Advances in Modeling Earth Systems, 5, 354–365, https://doi.org/10.1002/jame.20029, 2013.

Schwartz, M. J., Read, W. G., Santee, M. L., Livesey, N. J., Froidevaux, L., Lambert, A., and Manney, G. L.: Convectively injected water vapor in the North American summer lowermost stratosphere, Geophysical Research Letters, 40, 2316–2321, https://doi.org/10.1002/grl.50421,

750   2013.

Sioris, C. E., Malo, A., McLinden, C. A., and D'Amours, R.: Direct injection of water vapor into the stratosphere by volcanic eruptions, Geophysical Research Letters, 43, 7694–7700, https://doi.org/10.1002/2016GL069918, 2016a.

Sioris, C. E., Zou, J., McElroy, C. T., Boone, C. D., Sheese, P. E., and Bernath, P. F.: Water vapour variability in the high-latitude upper troposphere – Part 2: Impact of volcanic eruptions, Atmospheric Chemistry and Physics, 16, 2207–2219, https://doi.org/10.5194/acp-16-

2207-2016, 2016b.



Solomon, S., Rosenlof, K. H., Portmann, R. W., Daniel, J. S., Davis, S. M., Sanford, T. J., and Plattner, G.-K.: Contributions of stratospheric water vapor to decadal changes in the rate of global warming, Science (New York, N.Y.), 327, 1219–1223, https://doi.org/10.1126/science.1182488, 2010.

Stenchikov, G. L., Kirchner, I., Robock, A., Graf, H.-F., Antuña, J. C., Grainger, R. G., Lambert, A., and Thomason, L.: Radiative forcing from the 1991 Mount Pinatubo volcanic eruption, Journal of Geophysical Research: Atmospheres, 103, 13 837–13 857, https://doi.org/10.1029/98JD00693, 1998.

Stevens, B. and Bony, S.: Water in the atmosphere, Physics Today, 66, 29–34, https://doi.org/10.1063/PT.3.2009, 2013.

Tao, M., Konopka, P., Ploeger, F., Yan, X., Wright, J. S., Diallo, M., Fueglistaler, S., and Riese, M.: Multitimescale variations in modeled stratospheric water vapor derived from three modern reanalysis products, Atmospheric Chemistry and Physics, 19, 6509–6534, https://doi.org/10.5194/acp-19-6509-2019, 2019.

Tian, W., Chipperfield, M. P., and Lü, D.: Impact of increasing stratospheric water vapor on ozone depletion and temperature change, Advances in Atmospheric Sciences, 26, 423–437, https://doi.org/10.1007/s00376-009-0423-3, 2009.

Timmreck, C., Mann, G. W., Aquila, V., Hommel, R., Lee, L. A., Schmidt, A., Brühl, C., Carn, S., Chin, M., Dhomse, S. S., Diehl, T., English, J. M., Mills, M. J., Neely, R., Sheng, J., Toohey, M., and Weisenstein, D.: The Interactive Stratospheric Aerosol Model Intercomparison Project (ISA-MIP): motivation and experimental design, Geoscientific Model Development, 11, 2581–2608, https://doi.org/10.5194/gmd-11-2581-2018, 2018.

Toohey, M., Stevens, B., Schmidt, H., and Timmreck, C.: Easy Volcanic Aerosol (EVA v1.0): an idealized forcing generator for climate simulations, Geoscientific Model Development, 9, 4049–4070, https://doi.org/10.5194/gmd-9-4049-2016, 2016.

Wang, T., Zhang, Q., Kuilman, M., and Hannachi, A.: Response of stratospheric water vapour to CO2 doubling in WACCM, Climate Dynamics, 54, 4877–4889, https://doi.org/10.1007/s00382-020-05260-z, 2020.