# Peer review of "The Impact of Volcanic Eruptions of Different Magnitude on Stratospheric Water Vapour in the Tropics"

_Atmospheric Chemistry and Physics, 2020_

## Short Comment (SC1) · 6 Dec 2020

Thank you for posting this interesting study. It may be useful to make the link with the literature on stratospheric aerosol injection. In Boucher et al. (2017), we showed that stratospheric water vapour (SWV) responds to aerosol heating in the tropical tropopause layer. We also quantified the contribution of this fast adjustment to the effective radiative forcing by stratospheric aerosols. For 10 Tg S $yr^{-1}$ emitted as $SO_2$ at 17 km altitude, we found a radiative effect from SWV of +0.19 $Wm^{-2}$, which represents 7.2% of the ERF or 13.2% of the IRF (see Table 1 in Boucher et al., 2017). Note that the radiative effect from the stratospheric heating itself is larger. I suspect however

the heating due to stratospheric aerosols in our model to be too high.

Boucher, O., C. Kleinschmitt, and G. Myhre, Quasi-additivity of the radiative effects of marine cloud brightening and stratospheric aerosol injection, *Geophysical Research Letters*, 44, 11158-11165, doi: 10.1002/2017gl074647, 2017.

---

## Referee Comment (RC1) · Anonymous Referee #1 · 22 Dec 2020

Overview: This paper examines changes in stratospheric water vapour in the presence of volcanic aerosols. This is done through several model runs with varying amounts of SO2 introduced into a model. They find the input of water vapour into the stratosphere increases due to aerosol induced heating at the tropical cold point. They quantify radiative forcing, surface temperature changes, and changes to the water vapor annual cycle. Below I've listed a number of points the authors should consider in revision to best represent their results.

1) First sentence is rather convoluted. It says "Volcanic eruptions increase the stratospheric water vapour (SWV) entry via long wave heating through the aerosol layer in

the cold point region, and this additional SWV alters the atmospheric energy budget."
Why don't you say instead "Increases in the temperature of the tropical cold point region through heating by volcanic aerosols results in increases in the entry value of stratospheric water vapour and subsequent water vapour feedbacks." (or something like that) And, an question, what exactly are you referring to as long-wave heating? I think it should be made clear that there is near-IR solar and terrestrial long-wave heating going on.

2) Line 43/43 says: "reducing the "freeze trap" effect originating from the increasingly low temperatures and consequent loss of WV due to ice formation and fallout. The reduced freezing trap character enhances the entry of water vapour into the stratosphere." I don't think "reducing the freeze trap" is the appropriate way to express what's going on, and the phrase at the end of the first sentence and second sentence are excessively wordy and not entirely clear. Regardless of the temperature change, the freeze trap still exists, you've just effectively changed the set point by increasing the cold point temperature. Water values in the troposphere are still much larger than those in the stratosphere, and the fluctuations discussed here are on the same order as that induced by the seasonal cycle in cold point temperatures. How about saying instead "thereby increasing the stratospheric entry value of water vapour."?

3) Line 44-68: this needs to be edited for clarity, or even deleted, but definitely shortened. I believe the genereal point being made is "Although we understand the mechanism, internal variability and scarcity of observations has made this difficult to observe in practice."

4) Suggestion: rather than using the term "the indirect pathway" why not talk about "changes in SWV due to aerosol induced changes in tropical cold point temperature?"

5) General note....Does ACP allow references to papers "in preparation"? I would suggest not doing so, and instead include a supplement with the relevant information from the "in prep" manuscript.

6) I would recommend a reorganization...instead of going from AOD straight to TOA radiative imbalance, instead, progress through temperature, water vapour, then radiative imbalance and surface temperature.

7) To put runs in perspective, it would be useful to note how the simulations would compare to known volcanic eruptions. This manuscript describes 5 quantities of S input, ranging from 2.5 Tg to 40 Tg. Pinatubo estimates are ∼20 Tg of SO2 (or 10 Tg of S). This should be noted, in particular when discussing resultant cold point perturbations. (In particular when considering what would be observable over internal variability)

8) Line 183/184 says "The month of September was chosen as an example since it lies in the time frame within which the annual cycle of water vapour entry into the stratosphere is enhanced." What do you mean by the annual cycle ....is enhanced? Are you simply stating that is when you expect the maximum water vapour entry into the stratosphere, or are you implying something about the amplitude? Please revise to make clear.

9) Question, I assume it is not just total AOD that is relevant for cold point warming, but also the vertical distribution of aerosol. Could you provide a plot of the vertical aerosol distribution (perhaps for your extreme case) and discuss whether that is realistic for a volcanic eruption? You note an extreme warming for the 40 Tg case, however, what happens for the 10 Tg case (which would be akin to Mt Pinatubo)?

10) Line 187 states there is a downward shift of the CP with increasing sulfur. I do not see that in Figure 3. The CP location appears to be at 100 hPa for all cases, it is just the temperature value at 100 hPa that changes.

11) Line 195...I asked this before, but I would recommend noting what long wave radiation is warming the aerosol layer. Is it terrestrial radiation (from below) or absorption of near IR (from the sun)?

12) Paragraph starting with line 195: Here would be a good place to compare with what

really happened during Pinatubo (which I assume the 10 TgS case would most closely match)

13) Line 217: The annual cycle is not described as the tape recorder, the manifestation of the annual cycle of tropical SWV (as seen in the vertical profile) is the water vapor tape recorder.

14) Line 247/248 states "The season SON was chosen as it is the period of highest SWV values in the lower stratosphere." It would be more accurate to state that SON follows the period where the entry value of stratospheric water vapor is highest, or restrict your statement to "highest SWV values in the tropical lower stratosphere". This is likely not the case in the polar regions.

15) Discussion at 255 and rest of paragraph. Does your model keep track of overshooting ice? If so, is there any in the tropics? If the model doesn't keep track of it, and there is none in the tropics, then delete this discussion.

16) Line 262-264: The text states "However, the water vapour enters the tropics mainly in the inner tropical region and then spreads throughout the globe, leading to values lower than expected according to the Clausius Clapeyron equation." I question whether this is actually true, that water vapour (and by extension mass) entry into the stratosphere is "mainly" through the inner tropics (which I assume is defined here as 5N to 5S). You should be able to demonstrate this with your model output. 17) Line 283. . .this should be stated at the start of the paper.

18) What does this mean? "In particular the seasonal cycle of the tape recorder is more strongly amplified in the Mt. Pinatubo run in 1992. . ." Are you saying that the difference between the WV minimum and the WV maximum is larger in 1992 that for climatology? Or, or you saying that the maximum in 1992 is larger than climatology?

19) And, just a general terminology comment: Why do you call the entry of water vapour into the stratosphere through the cold point "indirect". That is pretty much how

all water enters the stratosphere, with overshooting convection not being predominant, and methane oxidation being the other significant source.

20) Line 378-383: Mote et al. used both SAGE II and MLS data. There is plenty of additional satellite data following the publication of the Mote et al. papers from which the annual cycle in the entry value of stratospheric water vapor can be calculated. Until you have compared with that whole body of data, I don't think you can say that the initial tape recorder analysis is biased such that the model output is correct.

21) Section 4.2: although the solar occultation satellite measurements immediately following Pinatubo were contaminated by the presence of aerosol, the microwave measurements should be OK (MLS water worked for 1.5 years), and there should be at least some sonde measurements (from FP and in situ aircraft instruments). However, instead of directly looking at WV, why don't you look at how the model compares for the tropical tropopause temperature anomalies. There should be good operational sonde data in addition to any satellite information from which to deduce an increase in CP temperature causing an increase in SWV. And, if SWV did increase, it should also be detected in the regular mid latitude frost point measurements done by NOAA at 40N.

22) Line 443: change "second poster" to "second post"
* * *

---

## Referee Comment (RC2) · Anonymous Referee #2 · 8 Jan 2021

Review of "The Impact of Volcanic Eruptions of Different Magnitude on Stratospheric Water Vapour in the Tropics" by Kroll et al.

Overall, this is an interesting paper on how volcanic eruptions perturb stratospheric water vapor. Âă I particularly like the novel data set —Âăa large ensemble of model simulations containing volcanic eruptions. Âă Overall, I judge this paper to be publishable after correction of a few points —ÂăI don't think these are hard, but they are important to address.

* The calculation of the forcing needs to be better explained. Âă In Solomon et al. (2010) and Dessler et al. (2013) papers, forcing was estimated using a fixed dynamical heat-

ing assumption (Fels et al. 1980), where it is assumed that the stratospheric circula-
tion remains fixed), so the temperature adjusts to maintain the heating rates.Âă This
is not what they're doing here.Âă They're letting temperatures adjust completely to
changes in SWV.Âă I think this is close to what was done in this paper (which should
probably be cited): Huang, Y., Zhang, M., Xia, Y., Hu, Y., & Son, S.-W. (2016). Is
there a stratospheric radiative feedback in global warming simulations? Climate Dy-
namics, 46(1), 177-186, doi:10.1007/s00382-015-2577-2 I should add that Huang,
Y., Wang, Y., & Huang, H. (2020). Stratospheric Water Vapor Feedback Disclosed
by a Locking Experiment. Geophysical Research Letters, 47(12), e2020GL087987,
doi:https://doi.org/10.1029/2020GL087987 found a very small net feedback when tem-
perature and water vapor are both allowed to adjust.Âă

In any event, they need to more completely discuss the fundamental assumptions that
go into the calculation of TOA forcing and how it is the same or different from what
previous investigators have done.

A related issue arises in Section 3.6. Here, they talk about forcing, but they're really
looking at flux differences at various altitudes. I don't think this is commonly referred to
as "forcing", which is normally a TOA quantity. This needs to be clarified.

* What is the cold point? A lot of analysis in the paper are of the temperature and
location of the cold point. They need to explain how they calculate that. Is this the
coldest temperature on one of the model's pressure levels, or are they doing some
kind of interpolation to find the lowest temperature, even if it is between the model
levels?

They also need to mention that the model has relatively poor vertical resolution around
the tropical tropopause. I'm not sure this will interfere with their conclusions, but it might
and they should probably mention this as an uncertainty in the paper.

* "However, the water vapour enters the tropics mainly in the inner tropical region and
then spreads throughout the globe, leading to values lower than expected according

to the Clausius Clapeyron equation." This statement and the analysis around it are built on a conceptual error. The process of entering the stratosphere and the process of dehydration are different and they can occur in different locations and on different time scales. Entry of air into the stratosphere occurs everywhere in the tropics were tropopause heating rates are positive. This is almost all of the tropics. Dehydration, on the other hand, tends to occur mainly over the deep tropics (Fig. 8 of Schoeberl, M. R., & Dessler, A. E. (2011). Dehydration of the stratosphere. Atmospheric Chemistry and Physics, 11(16), 8433-8446, doi:10.5194/acp-11-8433-2011). That's why you get such a good agreement vs. deep tropical temperatures.

One other note: I'm surprised that there's such good agreement between zonal average temperatures and water vapor. One paper that might be of interest is Oman, L., Waugh, D. W., Pawson, S., Stolarski, R. S., & Nielsen, J. E. (2008). Understanding the changes of stratospheric water vapor in coupled Chemistry-Climate Model simulations. Journal of the Atmospheric Sciences, 65(10), 3278-3291, doi:10.1175/2008jas2696.1, which shows that it is the temperatures in the limited dehydration region that really matter.

* code availability: The code and data availability statement does not actually tell the reader where the code and data can be downloaded from. I know that the MPI GE can be obtained from the ESGF, but where are the other data sets available? If they're not making these available, they should come right out and say it. Also, are they archiving their analysis code?

More minor comments: Water vapor is expressed in the paper as mass mixing ratio (ppmm). This is highly non-standard and makes comparison to the rest of the literature, which almost exclusively uses volume mixing ratio, harder than it needs to be. I would go through the paper and convert ppmm to ppmv.

In various places in the paper, such as on figure 3, it says "10ˆ6 per meter extinction".Âă Maybe I'm missing something, but that doesn't sound right —Âăshould it be 10ˆ-6?

[Figure]

The paper is already quite long.Âă I would suggest removing the paragraph beginning on line 94 ("In the following ...").Âă While I understand the desire to put this material here, let me assure you that no one actually reads these paragraphs.

The paper is not terribly written, but there are a lot of grammar errors. One frequent error was a missing comma after an introductory phrase (https://www.grammarly.com/blog/commas-after-introductory-phrases/). For example, on line 175, it says: As a consequence of the negative TOA imbalance the surface temperatures ... but there should be a comma after imbalance. That error is made repeatedly and should be fixed. In general, the paper would benefit from a close-read from a native english speaker.

Fig. 4 shows SWV at 100 hPa, but 100 hPa may not be in the stratosphere in the tropics.

The use of so many footnotes is unusual and seems odd. I guess I would encourage the authors to ask themselves whether this material really needs to be in a footnote. If it's sufficiently important, it should be in the main text. If not, perhaps it can be left out.
* * *

---

## Author Comment (AC1) · 14 Jan 2021

Thanks for your comment. As our study focuses on volcanic eruptions we primarily did not compare to geoengineering scenarios. However, we agree that going beyond volcanic eruptions would be of interest and added a citation of the respective work in our outlook.

---

## Author Response (AR2)

**Response To Reviews**

Clarissa Kroll

February 2021

**1 Editorial Review**

We thank the editor Slimane Bekki for his helpful comments. The corresponding changes made in the manuscript are listed below: the editor's questions/comments are in black, answers in grey and changes in the manuscript in parenthesis and blue.

1) Comments to the Author: Some additions to the text could be written better. Section 3.1: where is the verb in the first part of this sentence? "The cooling documented by satellite measurements from the microwave sounding unit (MSU) of -0.3 K between June 1991 and December 1995 after ENSO removal and reached a peak cooling of -0.5 K (Soden et al., 2002)."

We reformulated the above sentence:

"The average surface cooling between June 1991 and December 1993 is -0.26 K with a maximum cooling of -0.36 K in our simulations, whereas the cooling documented by satellite measurements from the microwave sounding unit (MSU) was -0.3 K between June 1991 and December 1995 after ENSO removal and reached a peak cooling of -0.5 K (Soden et al., 2002)"

2) Section 4.2: I could not find a Santer (2003) in the references. "Santer (2003) discuss a global warming of the lower stratosphere caused by the eruption of Mt Pinatubo. "

We apologize, Santer et al. (2003) is now listed in the references.

"Santer, B. D., Sausen, R., Wigley, T. M. L., Boyle, J. S., AchutaRao, K., Doutriaux, C., Hansen, J. E., Meehl, G. A., Roeckner, E., Ruedy,R., Schmidt, G., and Taylor, K. E.: Behavior of tropopause height and atmospheric temperature in models, reanalyses, and observations:Decadal changes, Journal of Geophysical Research: Atmospheres, 108, ACL 1–1–ACL 1–22, https://doi.org/10.1029/2002JD002258, https:785//agupubs.onlinelibrary.wiley.com/doi/abs/10.1029/2002JD002258, 2003"

3) Grammar: "The warming in the time frame of the Mt Pinatubo eruption as measured by MSU amountS to 0.75-1.5 K. " Angell (1997) reportS the stratospheric warming due the eruption of Mt. Pinatubo based on radiosonde data. The warmest seasonal anomaly in the 100-50 hPa layer at the equator REACHED (2.00.8) K, where we find an ensemble mean of 1.8 K for the 5 Tg S and 5.2 K for the 10 Tg S scenario in OND 1991. The corresponding SWV increase at background levels of 4.5-5 ppmv WAS ESTIMATED TO BE 1.13-1.3 ppmv, WHICH IS CONSISTENT with our findings (Figure 7)."

We corrected the sentences to read:

"Santer (2003) discuss a global warming of the lower stratosphere caused by the eruption of Mt Pinatubo amounting to 0.75-1.5 K as measured by MSU. Angell (1997) reports the stratospheric warming due the eruption of Mt. Pinatubo based on radiosonde data. The warmest seasonal anomaly in the 100-50 hPa layer at the equator reached (2.0±0.8) K, where we find 1.8 K for the 5 Tg S and 5.2 K for the 10 Tg S scenario in OND 1991. The corresponding SWV increase at background levels of 4.5-5 ppmv can be estimated to be 1.13-1.3 ppmv based on a 12 % increase of SWV per Kelvin. These values are consistent with our findings (Figure 7)."

4) L378-383 A bit too convoluted. make 2 sentences. "The slightly higher values reported for early 1990 fall into the era of the Mt. Pinatubo eruption, however, and may be increased compared to the multi- year mean due to an amplification of the seasonal signal by the presence of the aerosol layer, an

effect leading to maximum deviations of up to +/- 1.6 ppmv in the 10 Tg S scenario of the EVAens simulations."

Done:

The slightly higher values reported for early 1990 fall into the era of the Mt. Pinatubo eruption, however, and may be increased compared to the multiyear mean due to an amplification of the seasonal signal by the presence of the aerosol layer. This effect led to maximum deviations of up to $\pm$ 1.6 ppmv in the 10 Tg S scenario of the EVAens simulations.

5) Section 3.6: Please, give the error is absolute term, i.e. in K, instead of "The associated errors in average cold point temperature lie below one percent of the respective cold point temperature."

We now list the error in Kelvin:

The associated errors in average cold point temperature lie below $\pm$ 1.1 K.

6) Grammar: why this 'found'? remove "the" before "those". Replace "form" by "WV dependency" ""In the inner tropics the found specific humidity values agree nicely with the those from the Clausius Clapeyron equation and its approximated form of a 12 % per Kelvin. "

Done:

"In the inner tropics the simulated specific humidity values agree nicely with those from the Clausius Clapeyron equation and its approximated WV dependency of a 12 % increase of SWV per Kelvin."

We thank two anonymous reviewers for their valuable and helpful comments. We considered the recommendations carefully and made corresponding changes in the text. In the following the reviewers' questions/comments are in black, answers in grey and changes in the manuscript in parenthesis and blue.

**2 First Peer Review**

1) First sentence is rather convoluted. It says "Volcanic eruptions increase the stratospheric water vapour (SWV) entry via long wave heating through the aerosol layer in the cold point region, and this additional SWV alters the atmospheric energy budget." Why don't you say instead "Increases in the temperature of the tropical cold point region through heating by volcanic aerosols results in increases in the entry value of stratospheric water vapour and subsequent water vapour feedbacks." (or something like that) And, an question, what exactly are you referring to as long-wave heating? I think it should be made clear that there is near-IR solar and terrestrial long-wave heating going on.

Yes: with long-wave heating we are referring to terrestrial long-wave and near-IR solar heating, although the terrestrial long-wave heating is dominating. We adapted the text to clarify this point. The first sentence was reformulated to ensure a better readability :

"Increasing in the temperature of the tropical cold point region through heating by volcanic aerosols results in increases in the entry value of stratospheric water vapour (SWV) and subsequent changes in the atmospheric energy budget."
.... "This study will focus on the indirect volcanic entry mechanism. In contrast to the direct entry, it can act for months or even years after volcanic eruptions since it depends on the heating by the aerosol layer in the stratosphere and not on the eruption event itself. This indirect volcanic entry is caused by the terrestrial long wave and near IR solar heating by the volcanic aerosol layer which leads to increased cold point temperatures."

2) Line 43/43 says: "reducing the "freeze trap" effect originating from the increasingly low temperatures and consequent loss of WV due to ice formation and fallout. The reduced freezing trap character enhances the entry of water vapour into the stratosphere." I don't think "reducing the freeze trap" is the appropriate way to express what's going on, and the phrase at the end of the first sentence and second sentence are excessively wordy and not entirely clear. Regardless of the temperature change, the freeze trap still exists, you've just effectively changed the set point by increasing the cold point temperature. Water values in the troposphere are still much larger than those in the stratosphere, and the fluctuations discussed here are on the same order as that induced by the seasonal cycle in cold point temperatures. How about saying instead "thereby increasing the stratospheric entry value of water vapour."?

We removed the reference to the "freeze trap" and changed the wording to:

"Consequently, the saturation water vapour pressure at the cold point is increased, thereby reducing the loss of WV due to ice formation and fallout. This mechanism enhances the entry of WV into the stratosphere."

3) Line 44-68: this needs to be edited for clarity, or even deleted, but definitely shortened. I believe the general point being made is "Although we understand the mechanism, internal variability and scarcity of observations has made this difficult to observe in practice."

Yes, this is the essence of the paragraph. As we would like to give credit to earlier works and outline the present standing of the field, we would refrain from deleting it, but shortened the text where possible.

"In an early, idealized study, Joshi and Shine (2003) underlined the importance of the aerosol profile and corresponding heating in the tropopause region. Despite the mechanisms being known, its analysis is still complicated as internal variability and scarcity of observations has made it difficult to observe it in practice. Additionally, even if SWV increases were recorded, the data usage might be discouraged, as was the case for Mt. Pinatubo by SAGE II because discrepancies between different satellites could not be satisfactory explained (Fueglistaler et al., 2013).
The scarcity of observational data is also reflected in the quality of the available reanalysis products for SWV, the usage of which in general is discouraged in some papers (e.g, Davis et al., 2017) and which

sometimes do not implicitly account for the volcanic forcing at all (Diallo et al. (2017), Tao et al. (2019)). The latter problem was also indicated by Löffler et al. (2016) when discussing SWV increases simulated for the eruption of Mt. Pinatubo. Nevertheless, by performing a regression analysis using a trajectory model fed by reanalysis data, Dessler et al. (2014) identified a SWV peak partially overlapping with the aerosol optical depth (AOD) signal of Mt. Pinatubo. As the SWV increase occurred before the eruption and AOD increase, the question remained if the peak in the residual might instead be caused by another source of variability. Another possible issue in their analysis is that some of the effects modeled by the regressors are themselves influenced by volcanic eruptions, which may lead to the volcanic signal being attributed to a different source. An example are the increases of the Brewer Dobson circulation (BDC), which, in the case of a volcanic eruption, can be partially caused by the heating due to the aerosol layer. Tao et al. (2019) also undertook an indirect quantification of the SWV increase after volcanic eruptions via another regression analysis, explicitly accounting for volcanic source terms. They found a clear volcanic signal in the expected time frame, but the magnitude of the SWV increase differed strongly between the individual reanalysis data sources."

4) Suggestion: rather than using the term "the indirect pathway" why not talk about "changes in SWV due to aerosol induced changes in tropical cold point temperature?"

We called the entry mechanism indirect to put it in contrast to the "direct" entry mechanism as used by e.g. Sioris et al (2016) for the injection of water vapour within the volcanic plume. The term is defined in the introduction to describe the additional volcanically induced entry of water vapour from the troposphere to the stratosphere which rules out contributions of methane oxidation. We additionally discuss the importance of the atmospheric temperature profile in the introduction. However, we would refrain from using the rather lengthy formulation "changes in SWV due to aerosol induced changes in tropical cold point temperature" since it might distract from the main message we would like to convey in the respective sentence. In order to underline the role of the volcanic eruption, we now refer to the "indirect volcanic pathway".

The corresponding changes were made throughout the entire manuscript.

5) General note. Does ACP allow references to papers "in preparation"? I would suggest not doing so, and instead include a supplement with the relevant information from the "in prep" manuscript.

Yes, ACP explicitly asks to cite authors to include manuscripts which are not published. At `https://www.atmospheric-chemistry-and-physics.net/submission.html` (16.02.2021) ACP states:

**References                                                                                    [Back to top]**

Papers should make proper and sufficient reference to the relevant formal literature. Informal or so-called "grey" literature may only be referred to if there is no alternative from the formal literature. Works cited in a manuscript should be accepted for publication or published already. In addition to literature, data and software used should be referenced (citations should appear in the body of the article with a corresponding reference in the reference list). These references have to be listed **alphabetically** at the end of the manuscript under the **first author's name.** Works "submitted to", "in preparation", "in review", or only available as preprint should also be included in the reference list. Please do not use bold or italic writing for in-text citations or in the reference list.

By now, the corresponding manuscript has been submitted to the Journal of Geophysical Research. We changed the wording accordingly. The information necessary for the understanding of our manuscript is already included.

New reference: "Azoulay, A., Schmidt, H., and Timmreck, C.: The Arctic polar vortex response to volcanic forcing of different strengths, J. Geophys. Res.,submitted, 2020"

6) I would recommend a reorganization. . .instead of going from AOD straight to TOA radiative imbalance, instead, progress through temperature, water vapour, then radiative imbalance and surface temperature.

Section 3.1 is meant for orientation allowing the readers to compare the characteristics of the different EVAens simulations with their respective reference volcano. Discussing the atmospheric temperature and WV changes as suggested and then going back to radiative imbalance and surface temperature only to resume the discussion of atmospheric water vapour and temperature profiles might reduce the reading flow and seem redundant. Therefore we would prefer to keep the section as is.

7) To put runs in perspective, it would be useful to note how the simulations would compare to known volcanic eruptions. This manuscript describes 5 quantities of S input, ranging from 2.5 Tg to 40 Tg. Pinatubo estimates are 20 Tg of SO2 (or 10 Tg of S). This should be noted, in particular when discussing resultant cold point perturbations. (In particular when considering what would be observable over internal variability)

Although there are some mentions in the text of comparable eruption strengths, we agree with the referee that a general overview of historical eruptions comparable to those used in the EVAens would be of interest. We prepared a table (Table 1) listing different tropical eruptions which could have similar effects as the eruptions simulated with the EVA forcing generator, although it should be used with caution since we are applying synthetic aerosol profiles. The table can be found in section 2.1. With respect to the comparison to cold point temperatures which could be expected for the Mt Pinatubo eruption we refer to section 3.5 where the cold point warming in the MPI-ESM historical simulations for Mt Pinatubo are discussed. Additionally, we now also compare the TOA imbalance and surface cooling of the 5 Tg S and 10 Tg S cases, to observational values in section 3.1.. A comparison of temperature changes and corresponding water vapour anomalies in observations is now made in section 4.2.

section 2.1:

Table 1: List of tropical volcanic eruptions with location, eruption date, and estimated amount of emitted sulfur. The sulfur amounts in parentheses represent the best estimates.

| volcano | location | eruption time | emitted S [Tg] | rereference |
|---------|----------|---------------|----------------|-------------|
| Mt Agung | 8 °S, 115 °E | 17 Mar 1963 | 2.5-5 (3.5) | Timmreck et al. (2018) and references therein |
| El Chichón | 17 °N, 93 °W | 4 Apr 1982 | 2.5-5 (3.5) | Timmreck et al. (2018) and references therein |
| Mt Pinatubo | 15 °N, 120 °E | 15 Jun 1991 | 5-10 (7) | Timmreck et al. (2018) and references therein |
| Tambora | 8 °S, 117 °E | April 1815 | 15-40 (30) | Marshall et al. (2018) and references therein |

section 3.1:
"In addition to the EVAens results, the TOA imbalance and global surface temperature changes for the historical Mt Pinatubo eruption are shown in black in Figure 2. The global TOA imbalance, peaking at -2.4 Wm$^2$, compares favourably with the approximately -3 Wm$^2$ from Earth Radiation Budget Satellite observations (Soden et al., 2002) when considering the standard deviations of our ensemble. The average surface cooling between June 1991 and December 1993 is -0.26 K with a maximum cooling of -0.36 K, whereas the cooling documented by satellite measurements from the microwave sounding unit (MSU) was -0.3 K between June 1991 and December 1995 after ENSO removal and reached a peak cooling of -0.5 K (Soden et al., 2002)."

section 4.2.:
"During the Mt. Pinatubo period, observations of SWV are scarce, especially as solar occultation measurements done by HALOE and SAGE II suffered from aerosol interference during this time. Therefore, the data usage is discouraged (e.g.Fueglistaler et al. (2013)). Although Fueglistaler et al. (2013) mention "anomalously large anomalies" of SWV values shortly after the Mt. Pinatubo eruption in SAGE II data, they also warn that the measurements may be biased by aerosol artifacts since the SWV signal is not present in the HALOE data. However, in the Boulder balloon data a 1-2 ppmv increase of SWV at 24-26 km and 18-20 km levels is registered for 1992 (Oltmans et al., 2000). Most likely due to additional contribution from e.g. methane oxidation or ENSO signals, the maximum increase is slightly larger than the 1.0-1.2 ppmv shown for the tropical region in Figure 7 or the value of approximately 0.5 ppmv at 70 hPa at corresponding latitude (compare Appendix F1). Santer et al (2003) discuss a global warming of the lower stratosphere caused by the eruption of Mt Pinatubo amounting to 0.75-1.5 K as measured by MSU. Angell (1997) reports the stratospheric warming due the eruption of Mt. Pinatubo based on radiosonde data. The warmest seasonal anomaly in the 100-50 hPa layer at the equator reached (2.0$\pm$0.8) K, where we find 1.8 K for the 5 Tg S and 5.2 K for the 10 Tg S scenario in OND 1991. The corresponding SWV increase at background levels of 4.5-5 ppmv can be estimated to be 1.13-1.3 ppmv based on a 12 % increase of SWV per Kelvin. These values are consistent with our findings (Figure 7)."

8) Line 183/184 says "The month of September was chosen as an example since it lies in the time frame within which the annual cycle of water vapour entry into the stratosphere is enhanced." What do you mean by the annual cycle . . ..is enhanced? Are you simply stating that is when you expect the

maximum water vapour entry into the stratosphere, or are you implying something about the amplitude? Please revise to make clear.

We rephrased to

"The month of September was chosen as an example since it lies in the season of relatively large water vapour entry into the stratosphere due to a maximum cold point temperature in boreal autumn and winter."

9) Question, I assume it is not just total AOD that is relevant for cold point warming, but also the vertical distribution of aerosol. Could you provide a plot of the vertical aerosol distribution (perhaps for your extreme case) and discuss whether that is realistic for a volcanic eruption? You note an extreme warming for the 40 Tg case, however, what happens for the 10 Tg case (which would be akin to Mt Pinatubo)?

Yes, you are completely correct. We discuss the vertical aerosol distribution for the EVAens and also compare it to an aerosol distribution for Mt. Pinatubo based on satellite retrieval (Figure 10 and 11). We also present the corresponding cold point temperatures for the season SON, showing MPI-ESM simulations with the idealized EVAens and the Mt. Pinatubo volcanic forcing. The corresponding CP warming in the simulations of Mt. Pinatubo is larger than the warming in the 10 Tg S simulations. Please check section 3.5. for the corresponding results.

No changes were made since the aerosol distributions are already discussed in the manuscript.

10) Line 187 states there is a downward shift of the CP with increasing sulfur. I do not see that in Figure 3. The CP location appears to be at 100 hPa for all cases, it is just the temperature value at 100 hPa that changes.

We apologize, the resolution of Figure 3 was inappropriate to show this effect. In the revised manuscript we included an updated Figure 3 in which these effects are visible.

11) Line 195. . .I asked this before, but I would recommend noting what long wave radiation is warming the aerosol layer. Is it terrestrial radiation (from below) or absorption of near IR (from the sun)?

Please check the response to comment 1).

12) Paragraph starting with line 195: Here would be a good place to compare with what really happened during Pinatubo (which I assume the 10 TgS case would most closely match)

As previously mentioned, we now compare our simulation results with observational data throughout the manuscript. We show results from the historical Mt Pinatubo simulations in Figure 2, we discuss these records in section 3.1 instead of 3.2. The changes in stratospheric water vapour are discussed in relation to reanalysis data, and now observational data as well, in section 4.2 of the discussion. Please refer to comments 7 and 21 for further changes regarding this section.

13) Line 217: The annual cycle is not described as the tape recorder, the manifestation of the annual cycle of tropical SWV (as seen in the vertical profile) is the water vapor tape recorder.

We changed the wording accordingly.

"The manifestation of the annual cycle of the tropical SWV as seen in the vertical profile is often described as the tape recorder signal (Mote et al., 1996): The variations of the tropical cold point temperatures controlling the water vapour entry into the stratosphere via the saturation water vapour pressure is imprinted on the stratospheric water vapour as music is imprinted on a tape. This leads to an annual cycle of bands of high and low water vapour content propagating upwards in the stratosphere with the BDC."

14) Line 247/248 states "The season SON was chosen as it is the period of highest SWV values in the lower stratosphere." It would be more accurate to state that SON follows the period where the entry value of stratospheric water vapor is highest, or restrict your statement to "highest SWV values in the

tropical lower stratosphere". This is likely not the case in the polar regions.

According to your suggestion we rephrased to

"The season SON was chosen, as it follows the period where the entry value of stratospheric water vapor is highest, leading to the highest SWV in the tropical lower stratosphere."

15) Discussion at 255 and rest of paragraph. Does your model keep track of overshooting ice? If so, is there any in the tropics? If the model doesn't keep track of it, and there is none in the tropics, then delete this discussion.

Yes, overshooting ice is included in our model. For further information on the implementation, please refer to Möbis and Stevens, 2012 (https://doi.org/10.1029/2012MS000199), which we now also cite in our manuscript. The corresponding description of the parameterizations includes no regional constraints.

"In addition to water vapour, sublimated lofted ice can also contribute to the SWV leading to higher values than expected based on the saturation water vapour alone. Overshooting events are considered in our convection parameterisation (Möbis and Stevens, 2012). However, at the location of the cold point, this lofted ice would still be in the ice state and not accounted for in the specific humidity term."

16) Line 262-264: The text states "However, the water vapour enters the tropics mainly in the inner tropical region and then spreads throughout the globe, leading to values lower than expected according to the Clausius Clapeyron equation." I question whether this is actually true, that water vapour (and by extension mass) entry into the stratosphere is "mainly" through the inner tropics (which I assume is defined here as 5N to 5S). You should be able to demonstrate this with your model output.

We agree with the referee that this formulation was unfortunate and reformulated the paragraph as follows:

"In the inner tropics the simulated specific humidity values agree nicely with the values from the Clausius Clapeyron equation and its approximated form of a 12 % increase of SWV per Kelvin. Considering the simplification of taking the average cold point temperatures between $[-5,5]°$ latitude instead of the minimum cold point temperatures this agreement is surprisingly good. Oman et al. (2008), for example, only found good agreement when considering the minimum cold point temperatures in the tropics. As they were analyzing a band between $[-10,10]°$ latitude a factor contributing to the difference to our result could be that the main contribution to dehydration of air parcels during their horizontal motion in the vertical ascent takes place in the inner tropics (Schoeberl and Dessler, 2011), the region to which our study is restricted.
Consistent with this analysis is the stronger discrepancy of approximately 1.6 ppmv between the SWV values predicted using the Clausius Clapeyron equation and the SWV output by the model when averaging over the entire tropics (compare Fig. C1)."

17) Line 283. . .this should be stated at the start of the paper.

According to your comment 7) we added a corresponding table to the manuscript. Please refer to comment 7) for details.

18) What does this mean? "In particular the seasonal cycle of the tape recorder is more strongly amplified in the Mt. Pinatubo run in 1992. . ." Are you saying that the difference between the WV minimum and the WV maximum is larger in 1992 that for climatology? Or, or you saying that the maximum in 1992 is larger than climatology?

The SWV maxima are increased with respect to the climatology in all volcanically perturbed simulations. In this case we wanted to underline the seasonality of the SWV increase in the Mt. Pinatubo simulations which is not simulated to that extent in the EVAens.

We rephrased the corresponding sentence to:

"In particular, the SWV increases show a seasonality enhancing the tape recorder amplitude in the historical simulations which is not present to that extent in the EVAens simulations. Notable is also a

SWV increase of 1.2 ppmv in the Mt. Pinatubo simulations of 1992 exceeding the 1.0 ppmv of 1991."

19) And, just a general terminology comment: Why do you call the entry of water vapour into the stratosphere through the cold point "indirect". That is pretty much how all water enters the stratosphere, with overshooting convection not being predominant, and methane oxidation being the other significant source.

Please refer to the response to comment 4).

20) Line 378-383: Mote et al. used both SAGE II and MLS data. There is plenty of additional satellite data following the publication of the Mote et al. papers from which the annual cycle in the entry value of stratospheric water vapor can be calculated. Until you have compared with that whole body of data, I don't think you can say that the initial tape recorder analysis is biased such that the model output is correct.

We are aiming to compare with the years simulated with the MPI-ESM/EVAens and discuss the respective discrepancies. In no way we want to question the initial analysis of the tape recorder signal based on model results. However, since we agree with the comment that additional papers discussing the amplitude of the tape recorder signal in observational data would be of use, we expanded the discussion with a comparison to the SPARC Data Initiative multi-instrument mean (SDI MIM) at 100 hPa for the years 2005-2010 reported by Davis et al, 2017.

"In general, the annual cycle of the tropical tropopause temperatures accounts for a variation of the SWV content of +/- 1.4 ppmv around the mean background at 110 hPa in SAGE II data of the early 1990s (Mote et al, 1996) and +/- 1.0-1.3 ppmv based on the SPARC Data Initiative multi-instrument mean (SDI MIM) at 100 hPa in 2005-2010 (Davis et al, 2017). In the MPI-GE the variation in the SWV tape recorder signal at the same height reaches up to +/- 1.1 ppmv. This is in accordance with the SPD MIM and only slightly lower than the SAGE II data. The slightly higher values reported for early 1990 fall into the era of the Mt. Pinatubo eruption, however, and may be increased compared to the multiyear mean due to an amplification of the seasonal signal by the presence of the aerosol layer. This effect led to maximum deviations of up to $\pm$ 1.6 ppmv in the 10 Tg S scenario of the EVAens simulations."

21) Section 4.2: although the solar occultation satellite measurements immediately following Pinatubo were contaminated by the presence of aerosol, the microwave measurements should be OK (MLS water worked for 1.5 years), and there should be at least some sonde measurements (from FP and in situ aircraft instruments). However, instead of directly looking at WV, why don't you look at how the model compares for the tropical tropopause temperature anomalies. There should be good operational sonde data in addition to any satellite information from which to deduce an increase in CP temperature causing an increase in SWV. And, if SWV did increase, it should also be detected in the regular mid latitude frost point measurements done by NOAA at 40N.

Our study mainly focuses on idealized simulation experiments exploiting the possibility to have larger ensembles. We agree with the reviewer that next to a comparison to reanalysis data a discussion of corresponding observational data is of interest. We added more references to observational data sets: the Boulder Balloon data set for SWV increases and radiosonde measurements. Additionally, we discuss the recorded warming in the cold point region after the eruption of Mt. Pinatubo, which we translate into possible SWV increases. Please refer to section 4.2 and comment 7) for further details. A study focusing solely on observational records would be a nice follow up project.

22) Line 443: change "second poster" to "second post"

Done.

**3 Second Peer Review**

Review of "The Impact of Volcanic Eruptions of Different Magnitude on Stratospheric Water Vapour in the Tropics" by Kroll et al. Overall, this is an interesting paper on how volcanic eruptions perturb stratospheric water vapor. I particularly like the novel data set a large ensemble of model simulations containing volcanic eruptions. Overall, I judge this paper to be publishable after correction of a few points. I don't think these are hard, but they are important to address.

1) The calculation of the forcing needs to be better explained. In Solomon et al. (2010) and Dessler et al. (2013) papers, forcing was estimated using a fixed dynamical heating assumption (Fels et al. 1980), where it is assumed that the stratospheric circulation remains fixed), so the temperature adjusts to maintain the heating rates. This is not what they're doing here. They're letting temperatures adjust completely to changes in SWV. I think this is close to what was done in this paper (which should probably be cited): Huang, Y., Zhang, M., Xia, Y., Hu, Y., Son, S.-W. (2016). Is there a stratospheric radiative feedback in global warming simulations? Climate Dynamics, 46(1), 177-186, doi:10.1007/s00382-015-2577-2 I should add that Huang, Y., Wang, Y., Huang, H. (2020). Stratospheric Water Vapor Feedback Disclosed by a Locking Experiment. Geophysical Research Letters, 47(12), e2020GL087987, doi:https://doi.org/10.1029/2020GL087987 found a very small net feedback when temperature and water vapor are both allowed to adjust. In any event, they need to more completely discuss the fundamental assumptions that go into the calculation of TOA forcing and how it is the same or different from what previous investigators have done.
A related issue arises in Section 3.6. Here, they talk about forcing, but they're really looking at flux differences at various altitudes. I don't think this is commonly referred to as "forcing", which is normally a TOA quantity. This needs to be clarified.

We have expanded the description of the forcing calculation in section 2.2. to address the reviewer's concerns. As konrad is a 1D RCE model, it has a highly parameterized "circulation". This "circulation" does not redistribute water vapour or other gases but solely leads to an adiabatic cooling. In the stratosphere the BDC is described by an upwelling velocity term which indeed is fixed at a constant, nonzero velocity throughout the simulations. For the calculations of the adjusted forcing the temperatures above the convective top are allowed to change. Consequently, the dynamical heating is not fixed. We point this out in the rephrased version of section 2.2. and also added a citation of Fels et al. (1980). The methods of Huang et al, (2016) and those we employ in our calculations are different. For example, Huang et al. use a GCM, whereas we use a 1D RCE model, which entails large differences in the representation of circulation. We, therefore, do not cite Huang et al., (2016) in this context but add a citation of the paper by Hansen et al. (2005) giving the definition of stratospherically adjusted and instantaneous forcing which we are following in our manuscript. However, we added a reference to Huang (2020), mentioning the possible effects of tropospheric adjustments in the results section.

While it is indeed common to define the forcing at the TOA, there are also many studies which define it at the tropopause (e.g. Hansen, 2005). The choice of labelling in our diagram with the term "forcing" in our diagram was unfortunate when referring to the entire atmosphere. We changed the label to read "flux differences".

section 2.2.:
"The stratospherically adjusted clear sky radiative forcing, as defined by Hansen et al. (2005), originating from the increase of stratospheric water vapour due to the indirect volcanic pathway (i.e. via tropopause warming by the aerosol layer), is calculated using the 1D radiative convective equilibrium (RCE) model konrad (Kluft et al. (2019), Dacie et al. (2019)). Konrad is designed to represent the tropical atmosphere. It uses the Rapid Radiative Transfer Model for GCMs (RRTMG) and a simple convective adjustment that fixes tropospheric temperatures up to the convective top according to a moist adiabat, whereas the temperatures in the higher atmospheric levels are determined by radiative-dynamical equilibrium. Being a 1D model, konrad employs a highly parameterized "circulation", i.e. an upwelling term constant in time which only causes adiabatic cooling. As the temperature above the convective top can adjust, this does not mean that the dynamical heating is fixed (compare Fels et al.(1980)).
.....
Using both equilibrium states the adjusted SWV forcing is determined from the flux differences at the top of the atmosphere. The corresponding instantaneous forcing as defined by Hansen et al. (2005) is calculated as the difference of tropopause fluxes obtained without running the perturbed atmosphere into equilibrium."

section 3.6:
"If in addition to the stratosphere also the troposphere were allowed to adjust, part of this effect may be counterbalanced (Huang et al, 2020)."

2) What is the cold point? A lot of analysis in the paper are of the temperature and location of the cold point. They need to explain how they calculate that. Is this the coldest temperature on one of the model's pressure levels, or are they doing some kind of interpolation to find the lowest temperature, even if it is between the model levels? They also need to mention that the model has relatively poor vertical resolution around the tropical tropopause. I'm not sure this will interfere with their conclusions, but it might and they should probably mention this as an uncertainty in the paper.

We agree that more details should be given and added a paragraph providing details on how we define and determine the cold point at the first mention of the cold point in the results section (section 3.2). In some figures a coarser resolution field was used (Fig. 3, Fig. 4 and Fig. 5). We have replaced these with the higher resolution input data.

"In the following analysis the cold point is estimated as the lowest temperature in the TTL region lying on full pressure levels of model output remapped to a vertical spacing of 10 hPa in the tropical tropopause region. The associated errors in average cold point temperature lie below 1.1 K the respective cold point temperature. "

3) "However, the water vapour enters the tropics mainly in the inner tropical region and then spreads throughout the globe, leading to values lower than expected according to the Clausius Clapeyron equation." This statement and the analysis around it are built on a conceptual error. The process of entering the stratosphere and the process of dehydration are different and they can occur in different locations and on different time scales. Entry of air into the stratosphere occurs everywhere in the tropics were tropopause heating rates are positive. This is almost all of the tropics. Dehydration, on the other hand, tends to occur mainly over the deep tropics (Fig. 8 of Schoeberl, M. R., Dessler, A. E. (2011). Dehydration of the stratosphere. Atmospheric Chemistry and Physics, 11(16), 8433-8446, doi:10.5194/acp-11-8433-2011). That's why you get such a good agreement vs. deep tropical temperatures. One other note: I'm surprised that there's such good agreement between zonal average temperatures and water vapor. One paper that might be of interest is Oman, L., Waugh, D. W., Pawson, S., Stolarski, R. S., Nielsen, J. E. (2008). Understanding the changes of stratospheric water vapor in coupled Chemistry-Climate Model simulations. Journal of the Atmospheric Sciences, 65(10), 3278-3291, doi:10.1175/2008jas2696.1, which shows that it is the temperatures in the limited dehydration region that really matter.

Thanks for these references. We reformulated the corresponding section as follows:

"In the inner tropics the simulated specific humidity values agree nicely with those from the Clausius Clapeyron equation and its approximated WV dependency of a 12 % per Kelvin. Considering the simplification of taking the average cold point temperatures instead of the minimum cold point temperatures this agreement is surprisingly good. Oman et al (2008) for example only found good agreement when considering the minimum cold point temperatures in the tropics. As they were analyzing a band between [-10,10] latitude, a contributing factor to our result could be that the main contribution to dehydration of air parcels during their horizontal motion in the vertical ascent takes place in the inner tropics Schoeberl.2011 (Schoeberl and Dessler, 2011) to which our study was restricted. Consistent with this analysis is the stronger discrepancy of approximately 1.6 ppmv between the SWV values predicted using the Clausius Clapeyron equation and the SWV output by the model when averaging over the entire tropics (compare Fig. C1)."

4) code availability: The code and data availability statement does not actually tell the reader where the code and data can be downloaded from. I know that the MPI GE can be obtained from the ESGF, but where are the other data sets available? If they're not making these available, they should come right out and say it. Also, are they archiving their analysis code?

We have improved the information about code and data availability. All information necessary to reproduce the authors work is stored in a repository. The reformulated "code and data availability statement" is:

"Code and data availability: Primary data and scripts used in the analysis that may be useful in

reproducing the author's work are archived by the Max Planck Institute for Meteorology and can be obtained via https://pure.mpg.de/pubman/faces/ViewItemOverviewPage.jsp?itemId=item_3270686. A more comprehensive description of the EVAens is provided by Azoulay et al., submitted to JGR (2020). Further information was archived by the Max Planck Institute for Meteorology under http://hdl.handle.net/21.11116/0000-0007-8B38-E. The 1D RCE model konrad is available online under https://github.com/atmtools/konrad."

5) More minor comments: Water vapor is expressed in the paper as mass mixing ratio (ppmm). This is highly non-standard and makes comparison to the rest of the literature, which almost exclusively uses volume mixing ratio, harder than it needs to be. I would go through the paper and convert ppmm to ppmv.

The corresponding changes were made.

6) In various places in the paper, such as on figure 3, it says "10^6 per meter extinction". Maybe I'm missing something, but that doesn't sound right should it be 10^-6?

We corrected the corresponding labels.

7) The paper is already quite long. I would suggest removing the paragraph beginning on line 94 ("In the following ..."). While I understand the desire to put this material here, let me assure you that no one actually reads these paragraphs.

We removed the corresponding section.

8) The paper is not terribly written, but there are a lot of grammar errors. One frequent error was a missing comma after an introductory phrase (https://www.grammarly.com/blog/commas-after-introductory-phrases/). For example, on line 175, it says: As a consequence of the negative TOA imbalance the surface temperatures ... but there should be a comma after imbalance. That error is made repeatedly and should be fixed. In general, the paper would benefit from a close-read from a native English speaker.

We corrected the corresponding phrases and a native speaker reread the manuscript.

9) Fig. 4 shows SWV at 100 hPa, but 100 hPa may not be in the stratosphere in the tropics.

We agree 100 hPa is in the boundary region, we now show values at 70 hPa which clearly lies above tropopause and the cold point as indicated in Figure 6.

10) The use of so many footnotes is unusual and seems odd. I guess I would encourage the authors to ask themselves whether this material really needs to be in a footnote. If it's sufficiently important, it should be in the main text. If not, perhaps it can be left out.

The footnotes were removed where possible (9 out of 12).

**References**

[1] J. K. Angell. "Stratospheric warming due to Agung, El Chichón, and Pinatubo taking into account the quasi-biennial oscillation". In: *Journal of Geophysical Research: Atmospheres* 102.D8 (1997), pp. 9479–9485. DOI: `https://doi.org/10.1029/96JD03588`. eprint: `https://agupubs.onlinelibrary.wiley.com/doi/pdf/10.1029/96JD03588`. URL: `https://agupubs.onlinelibrary.wiley.com/doi/abs/10.1029/96JD03588`.

[2] Alon Azoulay, Hauke Schmidt, and Claudia Timmreck. "The Arctic polar vortex response to volcanic forcing of different strengths". In: *J. Geophys. Res., submitted* (2020).

[3] Sean M. Davis et al. "Assessment of upper tropospheric and stratospheric water vapor and ozone in reanalyses as part of S-RIP". In: *Atmospheric Chemistry and Physics* 17.20 (2017), pp. 12743–12778. DOI: `10.5194/acp-17-12743-2017`.

[4] A. E. Dessler et al. "Variations of stratospheric water vapor over the past three decades". In: *Journal of Geophysical Research: Atmospheres* 119.22 (2014), pp. 12, 588–12, 598. ISSN: 01480227. DOI: `10.1002/2014JD021712`.

[5] M. Diallo et al. "Significant Contributions of Volcanic Aerosols to Decadal Changes in the Stratospheric Circulation". In: *Geophysical Research Letters* 44.20 (2017), pp. 10, 780–10, 791. ISSN: 00948276. DOI: `10.1002/2017GL074662`.

[6] S. Fueglistaler et al. "The relation between atmospheric humidity and temperature trends for stratospheric water". In: *Journal of Geophysical Research: Atmospheres* 118.2 (2013), pp. 1052–1074. ISSN: 01480227. DOI: `10.1002/jgrd.50157`.

[7] Manoj M. Joshi and Keith P. Shine. "A GCM Study of Volcanic Eruptions as a Cause of Increased Stratospheric Water Vapor". In: *Journal of Climate* 16.21 (2003), pp. 3525–3534. ISSN: 0894-8755. DOI: `10.1175/1520-0442(2003)016{\textless}3525:AGSOVE{\textgreater}2.0.CO;2`.

[8] Michael Löffler, Sabine Brinkop, and Patrick Jöckel. "Impact of major volcanic eruptions on stratospheric water vapour". In: *Atmospheric Chemistry and Physics* 16.10 (2016), pp. 6547–6562. DOI: `10.5194/acp-16-6547-2016`.

[9] Benjamin Möbis and Bjorn Stevens. "Factors controlling the position of the Intertropical Convergence Zone on an aquaplanet". In: *Journal of Advances in Modeling Earth Systems* 4.4 (2012). DOI: `https://doi.org/10.1029/2012MS000199`. eprint: `https://agupubs.onlinelibrary.wiley.com/doi/pdf/10.1029/2012MS000199`. URL: `https://agupubs.onlinelibrary.wiley.com/doi/abs/10.1029/2012MS000199`.

[10] Philip W. Mote et al. "An atmospheric tape recorder: The imprint of tropical tropopause temperatures on stratospheric water vapor". In: *Journal of Geophysical Research: Atmospheres* 101.D2 (1996), pp. 3989–4006. ISSN: 01480227. DOI: `10.1029/95JD03422`.

[11] S. Oltmans et al. "Increase in stratospheric water vapor from balloon-borne, frostpoint hygrometer measurements at Washington, DC and Boulder, Colorado". In: *Geophysical Research Letters* 27.PreJuSER-44738 (2000).

[12] Luke Oman et al. "Understanding the Changes of Stratospheric Water Vapor in Coupled Chemistry Climate Model Simulations". In: *Journal of the Atmospheric Sciences* 65.10 (2008), pp. 3278–3291. DOI: `10.1175/2008JAS2696.1`. URL: `https://journals.ametsoc.org/view/journals/atsc/65/10/2008jas2696.1.xml`.

[13] B. D. Santer et al. "Behavior of tropopause height and atmospheric temperature in models, reanalyses, and observations: Decadal changes". In: *Journal of Geophysical Research: Atmospheres* 108.D1 (2003), ACL 1-1-ACL 1–22. DOI: `https://doi.org/10.1029/2002JD002258`. URL: `https://agupubs.onlinelibrary.wiley.com/doi/abs/10.1029/2002JD002258`.

[14] M. R. Schoeberl and A. E. Dessler. "Dehydration of the stratosphere". In: *Atmospheric Chemistry and Physics* 11.16 (2011), pp. 8433–8446. DOI: `10.5194/acp-11-8433-2011`. URL: `https://acp.copernicus.org/articles/11/8433/2011/`.

[15] Christopher E. Sioris et al. "Direct injection of water vapor into the stratosphere by volcanic eruptions". In: *Geophysical Research Letters* 43.14 (2016), pp. 7694–7700. DOI: `10.1002/2016GL069918`.

[16] Christopher E. Sioris et al. "Upper tropospheric water vapour variability at high latitudes – Part 1: Influence of the annular modes". In: *Atmospheric Chemistry and Physics* 16.5 (2016), pp. 3265–3278. DOI: `10.5194/acp-16-3265-2016`.

[17] Christopher E. Sioris et al. "Water vapour variability in the high-latitude upper troposphere – Part 2: Impact of volcanic eruptions". In: *Atmospheric Chemistry and Physics* 16.4 (2016), pp. 2207–2219. DOI: 10.5194/acp-16-2207-2016.

[18] Brian J. Soden et al. "Global cooling after the eruption of Mount Pinatubo: a test of climate feedback by water vapor". In: *Science (New York, N.Y.)* 296.5568 (2002), pp. 727–730. DOI: 10.1126/science.296.5568.727.

[19] Mengchu Tao et al. "Multitimescale variations in modeled stratospheric water vapor derived from three modern reanalysis products". In: *Atmospheric Chemistry and Physics* 19.9 (2019), pp. 6509–6534. DOI: 10.5194/acp-19-6509-2019.